# Benchmarking Vision Models Under Generative Continuous Nuisance Shifts

## Abstract

One important challenge in evaluating the robustness of vision models is controlling individual nuisance factors independently. While some simple synthetic corruptions are commonly applied to existing models, they do not fully capture all realistic and relevant distribution shifts of real-world images. To overcome this challenge, we apply LoRA adapters to diffusion models to realize a wide range of individual nuisance shifts in a continuous manner. While existing generative benchmarks perform manipulations in one step, we argue for gradual and continuous nuisance shifts, as they allow evaluating the sensitivity and failure points of vision models. With this in mind, we perform a comprehensive large-scale study to evaluate the robustness and generalization of various classifiers under various nuisance shifts. Through carefully-designed comparisons and analysis, we reveal multiple valuable observations: 1) More modern and larger architectures trained on larger datasets tend to be more robust to various nuisance shifts and fail later for larger scales. 2) Pre-training strategy influences the robustness and fine-tuning a CLIP classifier improves the standard accuracy but deteriorates the robustness. 3) The accuracy drops only account for one dimension of robustness and the failure point analysis should be considered as an additional dimension for robustness evaluation. We hope our continuous nuisance shift benchmark can provide a new perspective on assessing the robustness of vision models.

## 1 Introduction

Machine learning models are typically validated and tested on fixed datasets under the assumption of independent and identically distributed samples. However, this may not fully reflect the true capabilities and potential vulnerabilities of models when deployed in dynamic real-world environments. The robustness in out-of-distribution (OOD) scenarios is important in the real world. In safety-critical applications, decision-makers might be interested in how models perform under various specific nuisance shifts and severity levels. The term "nuisance shifts" refers to any intervention on a considered image distribution that alters the visual information while not changing the class of a considered target object, which can include the weather, style, or background.

In the past, various benchmarks have been proposed to evaluate the robustness of computer vision models. One line of benchmarks manually collects data with nuisance shifts [1, 12, 17, 18, 20, 34, 41, 45]. Yet, such approaches are not scalable and often include only a small variety of nuisance shifts. While Hendrycks and Dietterich [16] reports accuracy drops for various synthetic corruption

Submitted to the 38th Conference on Neural Information Processing Systems (NeurIPS 2024) Track on Datasets and Benchmarks. Do not distribute.

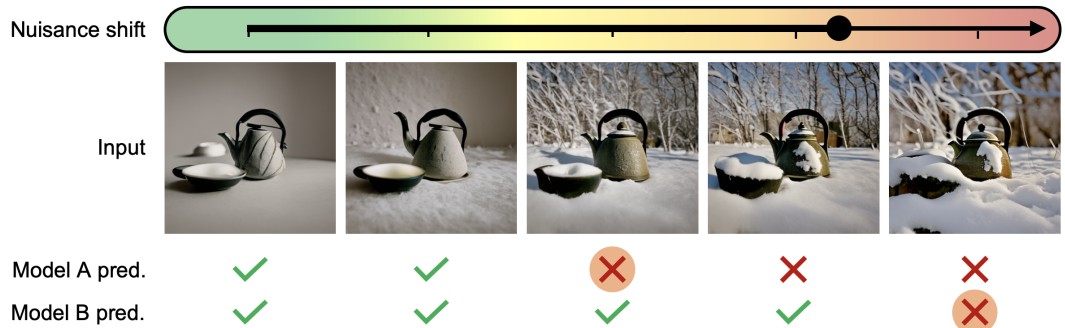

Figure 1: **Benchmarking Continuous Nuisance Shifts.** We find the *failure point* (highlighted in red) for different models under various nuisance shifts. This enables a fine-grained understanding of a model's robustness in various conditions.

types and levels of corruption, they are not always relevant in the real world and do not represent all real-world nuisance shifts.

On the other hand, synthetic datasets offer opportunities for evaluating deep neural networks. They allow the generation of various instances of a specific object class with specified context and nuisance shifts. While rendering pipelines allow precise control of several variables and are applied for benchmarking [3, 21, 23, 35], some nuisance shifts are hard to realize using traditional pipelines, such as weather variations like snow. Recent development in diffusion models has enabled the application of generative models for training [10, 15] and benchmarking vision models [29, 30, 40, 44].

However, all previous approaches define *binary* nuisance shifts by considering the existence or absence of that shift, which may contradict their continuous realization in real-world scenarios. For example, the snow level in an environment can range from light snowfall to objects fully covered with snow. While one model might fail at both levels, a different model might only fail when the object is heavily occluded. Thus, it is necessary to realize continuous shifts to evaluate the sensitivity of vision models and their failure points.

To overcome this shortcoming, we apply LoRA [19] adapters to realize a continuous variation of given nuisance shifts, and we use them for benchmarking a variety of classifiers along the following axes: (i) architecture, (ii) number of parameters, and (iii) pre-training and classification paradigms. Our new benchmark opens the path for robustness metrics beyond ImageNet accuracy: Evaluating on continuous levels allows computing the accuracy drop at specified scales and the failure point of models under a specific shift. In contrast to previous works that conduct analysis on two levels, our study reveals the following findings considering multiple levels of scales: 1) More modern and larger architectures are more robust to various nuisance shifts. 2) If a model is trained on more data using a classification or a surrogate loss, it is more robust independent of the standard accuracy. 3) Fine-tuning typically improves the standard accuracy. However, its impact on robustness varies depending on the considered models. 4) In addition to the accuracy drop as one measure of robustness, the point of failure might be a similarly important quantity to consider when the robustness with respect to a specific shift level is of relevance. Our results show that the two quantities are not always aligned and should be considered as two separate dimensions of robustness.

One essential requirement for using synthetic images for benchmarking is to ensure that the considered images correspond to the class distribution. Manually checking the quality of images is still common practice [44]. However, this does not allow scaling the analysis. Some approaches have been proposed for automated filtering, but there is no standard dataset for evaluating filtering strategies. We manually annotate a dataset with filter labels and use it to propose a filtering mechanism for removing out-of-class samples.

In summary, our work makes the following contributions: **(i)** We provide a framework for implementing and benchmarking vision models with respect to nuisance shifts under continuous severity levels.

(ii) We collect an annotated dataset for benchmarking out-of-class filtering strategies. We propose a novel filtering mechanism and apply it to our generated images. (iii) We evaluate the robustness of a variety of classifiers along different scales with respect to nuisance shifts with multiple scales. (iv) We publish a dataset for benchmarking the robustness of classifiers with respect to 14 diverse nuisance shifts at six severity levels. We additionally provide 1400 trained LoRA sliders that can be used for computing shift levels in a continuous manner.

## 2 Related work

**Robustness.** When referring to natural robustness, we consider the relative accuracy drop of a classifier with respect to interventions that alter images from a base distribution, building on the formalism introduced by Drenkow et al. [9]. While the robustness to generic distribution shifts is of interest, we consider the robustness with respect to specific nuisance shifts that can be modeled as causal interventions on the environment, the appearance, the object, or the renderer. We define such interventions in a continuous manner on a metric scale.

**Benchmarking Robustness.** Early approaches for benchmarking robustness and generalizability of models used fixed datasets [6, 7, 24], but this lacks scalability and fails to capture the failure points some models could face in real-world applications since they usually measure performances under the assumption of independent and identically distributed samples. To address this, a first line of research involves manually collecting data with nuisance shifts [1, 12, 17, 18, 20, 34, 41, 45].

However, these methods are often time-consuming and labor-intensive because they require data crawling and human annotations. Moreover, they usually capture only a subset of nuisance shifts that models may encounter in the real world and it is challenging to ensure the independence of these annotated nuisances. Additionally, it is possible to manually apply additional nuisances to evaluate their robustness in a more controlled manner, for example with image corruptions [16] or adversarial attacks [5, 31, 37]. The second line of research uses synthetic data for benchmarking, which offers the ability to generate a large and diverse range of nuisance shifts with precise control [3, 21, 35] but are limited to nuisance that can be easily modelled (*e.g.*, lighting, fog, occlusions). Recent developments in diffusion models have allowed some notable progress in the possibility of creating synthetic benchmark dataset [29, 30, 40, 44] with realistic data and more possibilities to control nuisances (*e.g.*, text-guided corruptions, counterfactual). In our work, we propose a framework for benchmarking vision models with respect to nuisance shifts under continuous severity levels, as well as a novel filtering mechanism for removing out-of-class samples from synthetic data.

## 3 Framework for Benchmarking

In this section, we present our methodology to realize continuous shifts for evaluating model's sensitivity with respect to such nuisance factors.

### 3.1 Continuous Nuisance Shifts for Benchmarking

For evaluating the robustness of image recognition models with respect to continuous scale nuisance shifts, two characteristics are desirable: (1) The severity of the considered shift can be controlled, allowing the estimation of the shift scale where a considered model fails. (2) Realizing a nuisance shift should not come along with factors of variations that might alter the class identity. The variations should be subtle and calibrated according to a pre-defined scale, allowing a fine-grained analysis on a distribution level when considering individual images.

**Methods for Realizing Continuous Shifts.** A natural way to realize nuisance factors are methods based on text prompts [25, 29, 40]. They follow the prompt template "A picture of a {class}" and "A picture of a {class} in {shift}". This, however, does not allow the gradual increase of a nuisance for a given image. In addition, the realized nuisance shift realized by the prompt addition "in {shift}" largely varies for different seeds and classes. The right figure in Fig. 3 illustrates that the nuisance

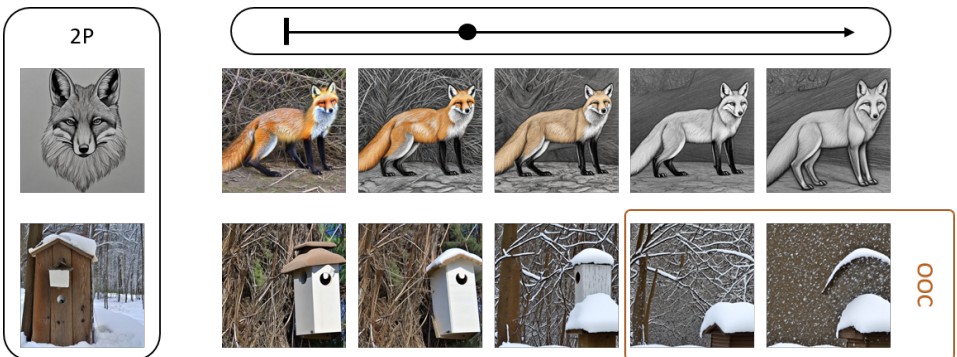

Figure 2: **Qualitative Examples for Prompt-Based and LoRA-Based Shifts including OOC Samples.** (1) We compare shifting using two text prompts (2P) and the LoRA strategy for one random seed. For 2P, the nuisance level is added in one step and the semantic structure clearly changes, while LoRA adapters allow a gradual variation. (2) One example sliding where the shifting strategy results in OOC samples for higher scales.

shift as measured by the difference of the CLIP [33] alignments of the base image and its shifted version to the prompt "A picture in snow" is dispersed. A qualitative example is given in Fig. 2 A naive approach for realizing continuous shifts involves computing the difference between two corresponding CLIP embeddings. We explored the naive strategy following the implementation of Baumann et al. [2], but we did not achieve robust nuisance shifts for a variety of classes. A different approach that allows realizing subtle variations involves LoRA [19] adapters. LoRA are low-rank matrices that can characterize the directions of nuisance shifts. Gandikota et al. [11] propose a strategy to learn concept sliders based on LoRA adapters to learn continuous concept variations. Similarly, we realize a nuisance shift by training a LoRA adapter that realizes a low-rank concept shift $s$ for a specific class $c$: $P_{\mathrm{GM}}(X|c+s) = P_{\theta_{\mathrm{SD}}}(X|c) \cdot P_{\theta_{\mathrm{LoRA}}}(X|c,s)$, where samples are drawn from the generative model (GM) by combining the pre-trained SD model with the learned LoRA adapter. We apply LoRA adapters that are learned based on concepts specified by language. As shown in Fig. 2, applying the LoRA slider allows realizing gradual nuisance shifts. We illustrate the average variation of the image and the realization of the shift for the LoRA approach and the approach based on two prompts (2P) in Fig. 3. The variation of the images is measured using the cosine similarity of the DINOv2-R class tokens of the base image and the shifted images, while the severity of the shift is measured using the text alignment to the prompt "A picture in snow". The LoRA adapter application allows gradual shifts, but the text-prompt-based application only allows one single scale for a given seed.

The variation of the number of noise steps [28] with active LoRA adapters controls to what extent the identity and semantics are modified when increasing the LoRA scale. We do not activate the LoRA adapter at earlier timesteps to realize variations that do not drastically change the semantic structure of the image since they are constructed at earlier timestamps of the diffusion process [27].

## 3.2 Accounting for the ImageNet Distribution

We aim to evaluate a model's robustness with respect to specific nuisance shifts $s$ that alter the base ImageNet distribution $p(X_{\mathrm{IN}}|c)$, which is conditioned on the 1k ImageNet classes $c$. For a more accurate estimate of the robustness with respect to a single considered shift, we desire a high model accuracy for the unshifted distribution. As pointed out by Vendrow et al. [40], the distribution of Stable Diffusion (SD) generated images $p(X_{\mathrm{SD}}|c)$ differs from the ImageNet distribution, resulting in lower classification accuracies.Therefore, we use the textual inversions provided by Vendrow et al. [40] to account for the ImageNet distribution and call it IN*: $p(X_{\mathrm{IN}*}|c) = p(X|c)$.

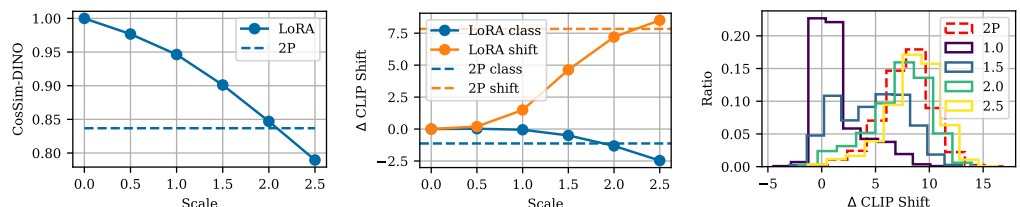

Figure 3: **Evaluation of Snow Sliding:** (1) Image variation is computed using the cosine similarity of DINOv2-R class tokens. (2) Computation of the shift measured by the CLIP difference of the base image and its shifted version. (3) Distribution of the applied shifts for various scales and 2P.

## 4 The Benchmarking Dataset

To evaluate filtering strategies for removing OOC samples, we collect a dataset. This section presents this dataset and the selected filtering strategy.

**Filtering of OOC Samples.** Current diffusion models allow the generation of diverse and realistic images $x \sim p(X|\mathbf{z})$ that are consistent with a desired condition $\mathbf{z} = [c, s_i]$ that involves the considered ImageNet class $c \in \mathbb{N} \mid 1 \leq c \leq 1000$ and the variable $s_i \in \mathbb{R}$ corresponding the level of a considered nuisance shift $i$. However, due to their probabilistic formulation, the generated sample might deviate from the the condition $\mathbf{z}$. While low-likely samples are in general not necessarily desired, long-tail samples also occur in the real world. For benchmarking applications, we are particularly concerned if the generated samples deviate from the original class $c$, i.e., the considered class cannot be characterized anymore. We call such samples "out-of-class" (OOC) samples [29]. Applying a LoRA adapter can leave the naturally learned manifold of the diffusion model and is, therefore, more prone to OOC samples (see Fig. 2). Evaluating the sensitivity to specific nuisance shifts requires removing the OOC samples generated by the shift's application. Therefore, we collect a dataset of generated images to evaluate the sliding process and strategies to automatically remove OOC samples.

**Dataset for Evaluating OOC Filtering Strategies.** To select a filter for detecting OOC samples, we collected a dataset for manual labeling: We pursue the following strategy:(i) In the first stage, 24k images are generated for 20 seeds, 5 LoRA scales, and 2 shifts per class for 100 random ImageNet classes in total. We select two very different shifts: One shift corresponds to a natural variation (snow), and the second shift corresponds to a style shift (cartoon style). (ii) Since we aim to find OOC samples that arise due to the application of the LoRA adapters, we remove all start samples without any shift that are low-likelihood samples, *i.e.* have a low text-alignment, and that are not classified as the corresponding class by multiple classifiers. After removing hard starting samples, the labeling dataset consists of around 18k images. (iii) To reduce the labeling effort, we filter out all easy samples that are (1) correctly classified by DINOv2-R and (2) one out of three classifiers (ResNet-50, DeiT-B/16, or ViT-B/16). (3) An additional requirement such that a sample is considered easy is a sufficiently high text alignment. (iv) Each hard image is labeled by two human annotators. To increase the dataset quality, we include soft labels if the image partially includes some characteristics of the class. So, each annotator can choose from the labels 'class', 'partial class properties', and 'not class'. An image is defined as OOC sample if at least one annotator considers the image as an OOC. For the remaining samples, an image is considered IC (in-class) if at least one annotator labeled the image a clear sample of the corresponding class. All details on the labeling strategy and the dataset statistics are found in Appendix A.

**OOC Filtering Strategy.** A filter serves its purpose if it removes all OOC samples, corresponding to a high true positive rate (TPR), while not removing too many in-class samples, which corresponds to a low false positive rate (FPR). Instead of simply applying a CLIP threshold as in Vendrow et al. [40], we consider a combinatorial selection approach, which requires two out of four detectors to be active. (i-ii) First, we consider text alignment to 'a picture of a {class}' and to 'a picture of a {class} in {shift}' computed via CLIP. (iii-iv) Additionally, we consider the cosine similarity to the starting images using the CLIP image encoder and the class tokens of DINOv2-R.For (i) and (ii), we select

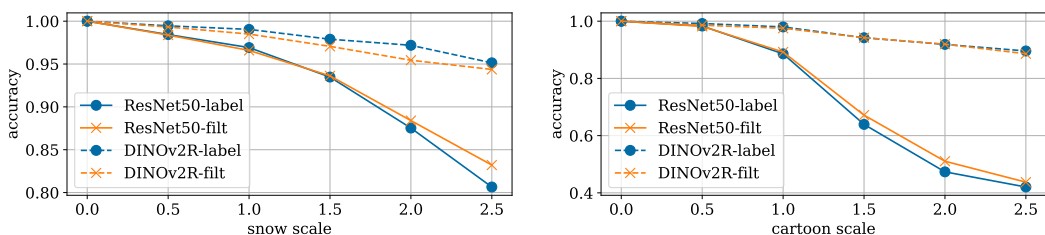

Figure 4: **Classification Accuracies on the Labeled and the Filtered Dataset.** The accuracy curves of a ResNet-50 and DINOv2-based classifier are comparable, which validates automatic filtering. We provide more results for more classifiers in Fig. 6.

the filtering thresholds such that 90% of the labeled OOC samples are removed. We do not require the detection of all OOC samples since ImageNet includes some class ambiguities. The threshold is selected in accordance with the highest achieved accuracies of classifiers on ImageNet [36, 42, 43]. The selected filter reaches a TPR of 87.9% and a FPR of 12.0% with an accuracy of 88.0%, while the simple CLIP-based thresholding reaches a TPR of 89.9% and a FPR of 35.7% with an accuracy of 65.1%. While being mostly effective, the filtering mechanism does not remove all OOC samples. Therefore, we plot the classification accuracy of DINOv2-R and ResNet-50 for the labeled and the filtered version in Fig. 4. These results show that the filtered dataset results in comparable accuracy drop as the labeled dataset for both considered shifts.

# 5 Benchmark

In this section, we discuss our benchmark. We present the evaluations on the OOD-CV dataset and the large scale analysis of ImageNet classifiers.

## 5.1 Evaluation on OOD-CV dataset

To measure the robustness, Zhao et al. [45, 46] introduce a benchmark dataset (OOD-CV) that includes out-of-distribution examples of then object categories for five different individual nuisance factors (*e.g.*, weather) on real data. OOD-CV is the only real-world dataset that provides accurate labels of various individual nuisance shifts. However, it only provides the coarse label *weather* for all weather-related nuisances instead of fine-grained labels such as *rain*, *snow*, *fog* or *other*. Following a similar approach in Sec. 4, we assign the fine-grained label using CLIP similarity. We detail the strategy for annotating OOD-CV using CLIP similarity and provide visualizations in Appendix A. We evaluate classifiers on both benchmarks. Specifically, we first train different classifiers (*i.e.*, ResNet-50, ViT, and DINO-v2-ViT) on the training set of the OOD-CV benchmark. We then evaluate their performance on the data generated using our approach. Besides, we also evaluate their performance on the OOD-CV benchmark for each annotated sub-nuisance independently. As shown in Fig. 5, the accuracy remains more or less constant with an accuracy around $95\%$ up to a nuisance scale of $1.5$. From $2.0$, the accuracy starts dropping, with the nuisance of *fog* and *sand* having the biggest impact. The resulting accuracy is consistently worse or similar to the accuracy of the highest nuisance scale of our generated data for the corresponding nuisance. We hypothesize that the bigger drop is due to a major limitation of the OOD-CV benchmark dataset: the nuisances are not completely disentangled, and part of the accuracy drop originates from various other factors (*e.g.*, image quality, image size, and noise). Another hint confirming that hypothesis is the slight accuracy increase (up to $+2.5\%$) for the *rain* and *snow* nuisances when increasing the nuisance scale from $0.0$ to $1.5$. Given that the models were trained on OOD-CV benchmark training set, and evaluated on our generated data. Thus, when corrupting the data with *snow* or *rain*, which closely relates to noise or pixelation from zooming in, the data becomes closer to the training data of the OOD-CV benchmark. Hence, the OOD-CV benchmark does not fully disentangle the annotated nuisances. In contrast, our approach allows for fine-grained control of nuisances, for a more complete understanding of a model's capability.

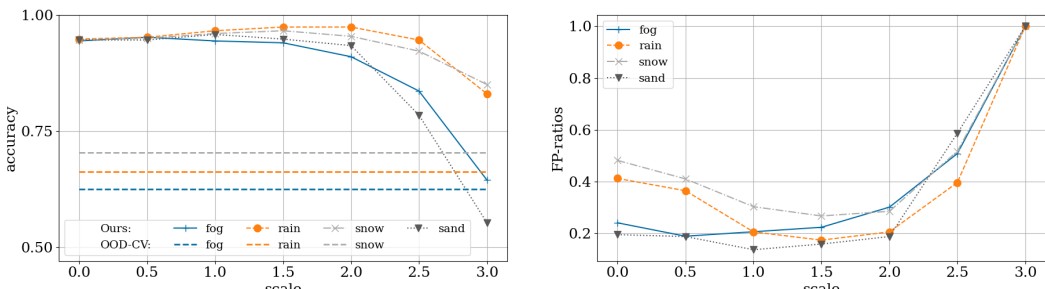

Figure 5: **Accuracies and Failure Point Ratios for the OOD-CV Benchmark.** The continuous scale nuisance shifts allow identifying the failure points of the models, while the OOD-CV dataset only provides the accuracy drop: horizontal lines show the average score for each sub-nuisance of the OOD-CV test dataset.

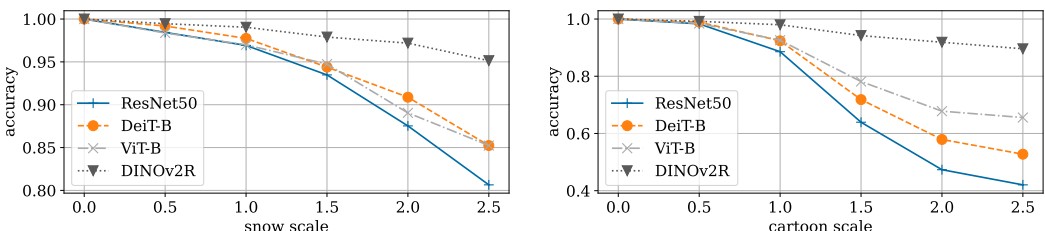

Figure 6: **Accuracies on the Labeled Dataset for Snow and Cartoon Shifts.** The accuracy drops on the labeled dataset showcase that various classifiers have varying sensitivities on different shifts.

## 5.2 Evaluated Models and Experimental Setup

We use our benchmark to evaluate the models along the following axes:

(i) *Architecture.* To compare architectures with a comparable number of parameters, we consider ResNet-50 [13], ViT-B/16 [8], DeiT-B/16 [38], DeiT-3-B/16 [39], and ConvNeXt-B [26]. All models are trained in a supervised manner.

(ii) *Model Size.* For ViT, we consider the small, medium, base, large, and huge variants of DeiT-3. For CNN, we consider the ResNet variants, *e.g.*, 18, 34, 50, 101, and 152.

(iii) *Paradigm and Training Data.* The selection of the training paradigm and the amount of training data are highly coupled. Therefore, we evaluate a set of models that differ with respect to the used data as well as their pre-training and classification strategy. We compare two supervised models: One model trained on IN1k, and the other model trained on IN21k and then fine-tuned on IN1k. To evaluate the effect of learning strategies, we include two more models that are trained on IN1k: A masked autoencoder (MAE) [14] and DINOv1 [4]. Additionally, we also include a VLM-based classifier using a pre-trained CLIP-model [33] and DINOv2 [32]. We include the zero-shot variant of CLIP and a version that is fine-tuned on IN1k. All models use ViT-B/16 as the backbone. Furthermore, we evaluate a diffusion classifer [22] on a smaller subset.

**Implementation Details.** As pointed about in Sec. 3.2, we use textual inversions to account for the ImageNet distribution. To evaluate the relance of this approach, we generate 200 images of 100 randomly selected ImageNet classes using standard SD2.0 and SD2.0 with the textual inversions of IN*. To illustrate the distribution gap, we compute the accuracies for ResNet-50 and DeiT. They achieve an accuracy of 68.2% and 71.6% for the SD distribution and 74.1% and 79.1% for the IN* distribution, which equals an accuracy drop of 6% and 8%, respectively. We perform all the following experiments using the IN* distribution. We use SD2.0 and we activate the LoRA adapters for the last 75% of noise steps. Due to the computational complexity, we perform sliding for 100 classes. To get an estimate of the robustness on a scale of ImageNet, we classify 1k classes using off-the-shelf classifiers without applying masking, as *e.g.*, done by Hendrycks et al. [17]. We ablate in Appendix A how the number of classes influences the robustness evaluations.

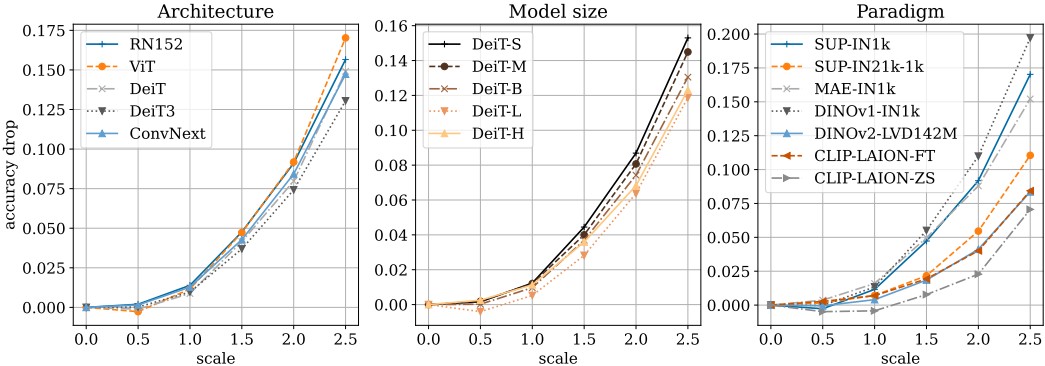

(a) Accuracy drops averaged over all considered shifts. Architecture (*left*): Models with the same training data and similar size. Model size (*middle*): The same model (DeiT) with different numbers of parameters. Paradigm (*right*): Supervised, self-supervised (MAE, DINOv1, and DINOv2-R), VLM (CLIP), all using ViT-B/16.

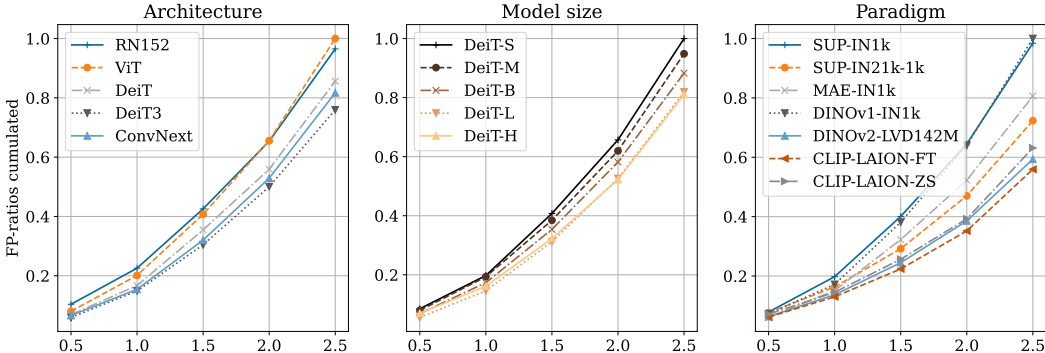

(b) Cumulative failure point rates: For each sliding trajectory that contains a failure sample, we sum the number of samples that were wrongly classified at a specific scale and apply a cumulative sum.

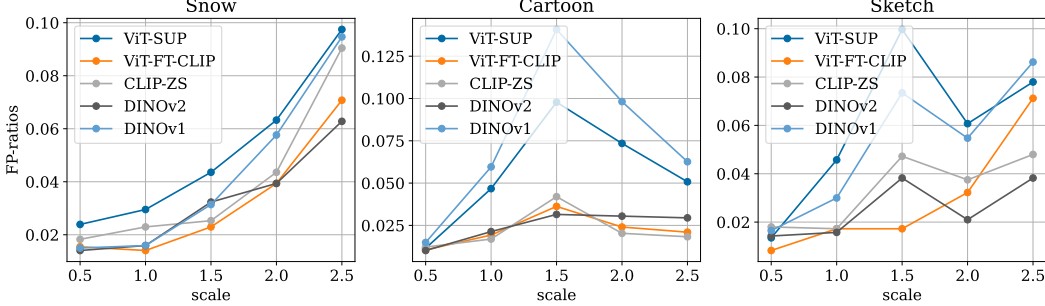

(c) Ratio of failure points per scale for various models and shifts: The distribution allows inferring at which scales various models fail most often.

Figure 7: **Benchmarking Classifiers and Shifts.** The visualization of accuracy drop and the distribution of failure points is provided for all shifts and the three considered axes.

Our filtering mechanism removes some samples along the sliding trajectory, *i.e.*, some seeds only include images from lower scales. To account for balanced dataset, we only evaluate the models for seeds that still contain all scales.

## 5.3 Analysis & Findings

Following Hendrycks et al. [17], we report the accuracy drop for 5 scales and 14 diverse shifts as a measure of robustness in Fig. 7a and the distribution of failure points in Fig. 7b and Fig. 7c. We list the shifts and more evaluations in Appendix A and discuss the findings in the following.

**More modern architectures improve the robustness even when using the same training data:** In our benchmark, DeiT3 achieves the highest robustness, while ConvNeXt and DeiT reach a similar performance. Interestingly, ResNet-152 is more robust than the standard ViT variant (Fig. 7a, Arch.).

**ConvNeXt fails later than ViT and ResNet-152:** The cumulated number of failure points in Fig. 7b is mostly consistent with the observations of the accuracy drops. However, we identify the following learnings when performing the failure point evaluation: While the accuracy drop did not allow to clearly differentiate the performance between ViT and DeiT, the failure mode-based evaluation shows a significantly better performance of the ConvNeXt model (Fig. 7b, Arch.). Similarly, ConvNeXt fails later than ResNet-152.

**Larger models are more robust:** This follows the results in Hendrycks et al. [17]. Our analysis shows that this behavior can be consistently reported for varying shift severities and for all considered nuisance factors (Fig. 7a, Model size). For this axis, the evaluation of the failure point is in line with the accuracy drop (Fig. 7b, Arch.).

**Using more data improves robustness:** The most robust classifiers were trained on large datasets, such as the CLIP models on LION or DINOv2 on LVD-142M. We report a better robustness for the model that was pre-trained on IN21k as well (Fig. 7a, Paradigm).

**MAE is the most robust pre-training strategy:** When comparing the models trained on the same dataset size, we observe that the fine-tuned MAE achieves the best robustness. (Fig. 7a, Paradigm) We use the DINOv1 model with a linear head for classification. Interestingly, it has a lower robustness than the ViT that was trained using a supervised loss. This might be attributed to the lower performance when only using linear probing. E.g., while the supervised approach (SUP-IN1k) showed better performance (Fig. 7a, Paradigm) than the MAE-based approach, MAE fails in average later than SUP-IN1k in case it fails (Fig. 7b, Paradigm).

**Some models have a larger accuracy drop but fail later.** Failure points are therefore a reasonable additional metric to evaluate the robustness of models with respect to continuous shifts.

**Fine-tuning improves the accuracy but deteriorates the robustness for CLIP:** The CLIP classifier applied in a zero-shot manner is more robust (Fig. 7a, Paradigm) while having a lower average accuracy: 89.5% vs. 84.2%. We report all accuracies in Appendix A.

**Diffusion classifiers seem not to be more robust than discriminative models.** We evaluate the accuracy drop of the DiT-based diffusion classifier for 1k images on a subset of our dataset (around 400 images) for the snow and the cartoon style shift due to computational constraints. When comparing the performance on the same reduced dataset, the accuracy drops for the LoRA scale 2 of snow (cartoon) shift by around 0.12 (0.37) percent points for the diffusion classifier using the L1 loss computations strategy [22] and by around 0.12 (0.30) percent points for a ViT-B model trained on IN1k. The accuracy drops reported for the evaluated discriminative models on the subset are almost in line with the experiments on the labeled dataset Fig. 6. We provide more results in Appendix A.

**Failure points differ across different types of shifts:** Comparing the failure point of various models largely differs when considering individual shifts as shown in Fig. 7c. Snow can be considered as an example shift that slightly changes the appearance and mainly adds a disturbance factor in the image. While there are some differences, the qualitative distribution is comparable for all models. On the contrary, the cartoon and sketch variation correspond to a style shift. Here, the failure points of less robust models are more concentrated.

## 6   Conclusion

This work fills the gap in generative robustness benchmarks that did not allow the application of a continuous shift level. In addition, we introduced the concept of failure points for benchmarking, providing an additional dimension to measure robustness. We applied LoRA adapters to realize fine-grained alterations of the image and benchmarked various classifiers along three axes. Furthermore, we discussed the importance of detecting out-of-class class samples when benchmarking using diffusion-generated images. We hope our proposed benchmark can motivate further research in the domain of using generated images for evaluating the natural robustness of vision models. Future work can improve the calibration and composition of various nuisance shifts.

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
