# A Appendix

This appendix provides supplementary information that is not elaborated in our main paper: We will discuss more details about the benchmarking dataset, the filtering, and image generation strategy. Additionally, we will provide more results. The supplementary material, including the benchmarking dataset, the filtering dataset, and the code to reproduce training, generation, and benchmarking, is provided in a GoogleDrive [1].

## A.1 Benchmark Details

This section provides more details about the benchmarking dataset. We first discuss the presented metrics. We then provide examples of the dataset and its distribution.

### A.1.1 Access to Benchmarking Dataset

We provide the number of images per shift after filtering in Fig. 10. The dataset contains $192,168$ images in total, with $32,028$ images per scale. We share all images on Google Drive in the folder *benchmarking_dataset*. Additionally, we add the anonymized metadata, including the annotations, as a JSON file. We will use these annotations to follow the MLCroissant standard once we publish the data on our own servers to allow easy and standardized loading of the dataset.

### A.1.2 Elaboration of Metrics

**Evaluation of Sliding.** We measure the delta CLIP shifts in Fig. 3 by computing the difference of the text alignments of the reference image and the slided image with the considered scale $s$: $\Delta\text{CLIP}_{\text{class}}(I_0, I_s) = \text{CLIP}_{\text{class}}(I_s) - \text{CLIP}_{\text{class}}(I_0)$ and $\Delta\text{CLIP}_{\text{shift}}(I_0, I_s) = \text{CLIP}_{\text{shift}}(I_s) - \text{CLIP}_{\text{shift}}(I_0)$, where the text-alignment to the class is computed via the text prompt "`A picture of a {class}`" and the text-alignment to the shift is computed via "`A picture in {shift}`". While the alignment to the shift is increasing, the alignment to the class is slightly decreasing.

**Failure Point.** In this work, we motivate the application of the failure point metric. In the following, we further discuss its computation and value. The failure point computation does not involve the relative number of failure cases. It only depicts the distribution of errors over various scale values and, therefore, considers a different dimension of robustness. We differentiate two ways of visualizing:
(1) Plotting of the failure point distribution as depicted in Fig. 7c: The reported values are divided by the total number of failure points of all considered models. The errors are not reported for scale 0, only depicting errors due to style shifts.
(2) Plotting of the cumulative failure point distribution as depicted in Fig. 7b: To better compare the number of images wrongly classified at a specific scale, we plot the cumulative distribution that reaches 1 for the largest scale, *i.e.*, indicating that all failed samples have failed the latest at the largest scale.

### A.1.3 List of Shifts and Example Images

The results are averaged over the following 14 shifts: cartoon style, plush toy style, pencil sketch style, painting style, design of sculpture, graffiti style, video game renditions style, style of a tattoo, heavy snow, heavy rain, heavy fog, heavy smog, heavy dust, and heavy sandstorm (see examples in Fig. 8 and Fig. 9). We train the sliders using the prompt template "`A picture of a {class} in {shift}`".

## A.2 Benchmarked Models

We present an overview of evaluated models in Tab. 1. It does not include all evaluated models. We refer to Tab. 2 for a complete list of all evaluated models.

---

[1] https://drive.google.com/drive/folders/1ZTbCwrpedcJ3tGS6U5C4NgnGI4PD1qBH?usp=sharing

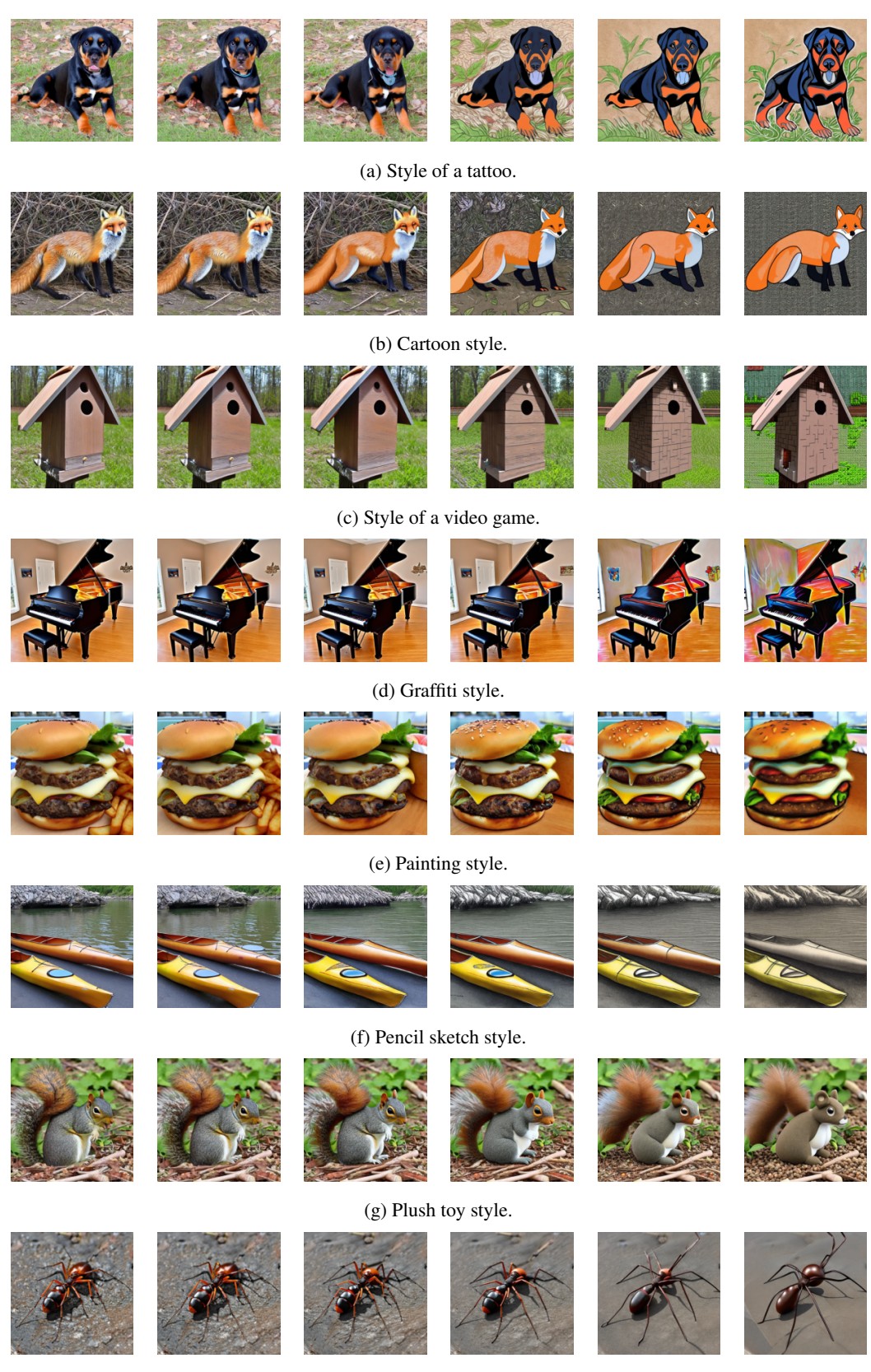

(a) Style of a tattoo.

(b) Cartoon style.

(c) Style of a video game.

(d) Graffiti style.

(e) Painting style.

(f) Pencil sketch style.

(g) Plush toy style.

(h) Design of a sculpture.

Figure 8: **Example sliding for various nuisance shifts.** We visualize six generated images with the corresponding scales as 0, 0.5, 1, 1.5, 2, and 2.5.

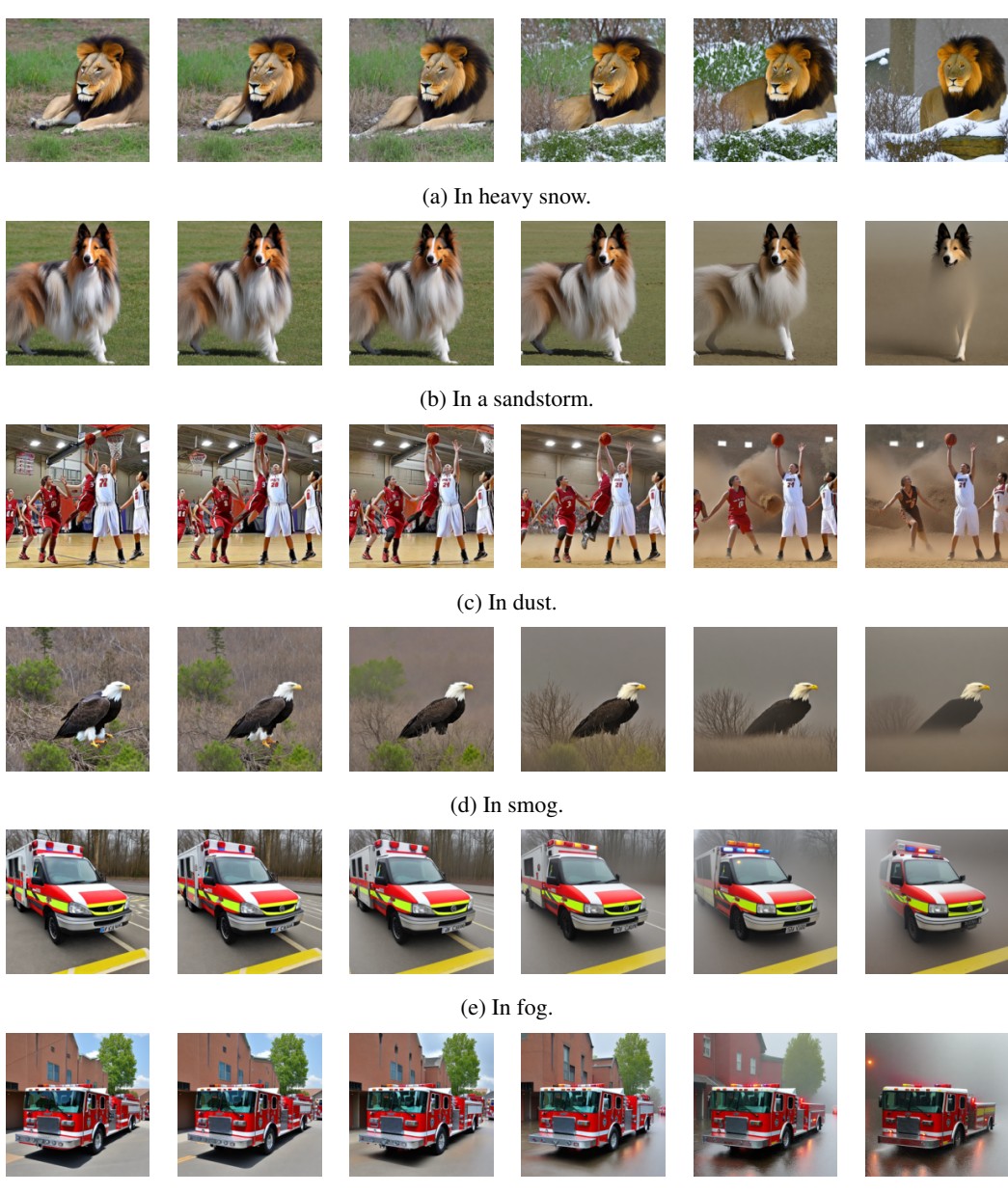

(a) In heavy snow.

(b) In a sandstorm.

(c) In dust.

(d) In smog.

(e) In fog.

(f) In heavy rain.

Figure 9: **Example sliding for various nuisance shifts.** We visualize six generated images with the corresponding scales as 0, 0.5, 1, 1.5, 2, and 2.5.

### A.3 More Results

We provide a table of accuracies and accuracy drops for all evaluated models and scales and the average accuracy and accuracy drop in Tab. 2. Additionally, we provide the failure point distribution for all evaluated models in Tab. 3. As discussed in the main paper, we also provide the results for the ResNet family in Fig. 11. Similar to the observations in Tab. 2, larger models result in a lower accuracy drop. We provide functionality to load the classification results for all images of the dataset in the shared code. All results are computed in a standardized way using the *easyrobust* [11] framework.

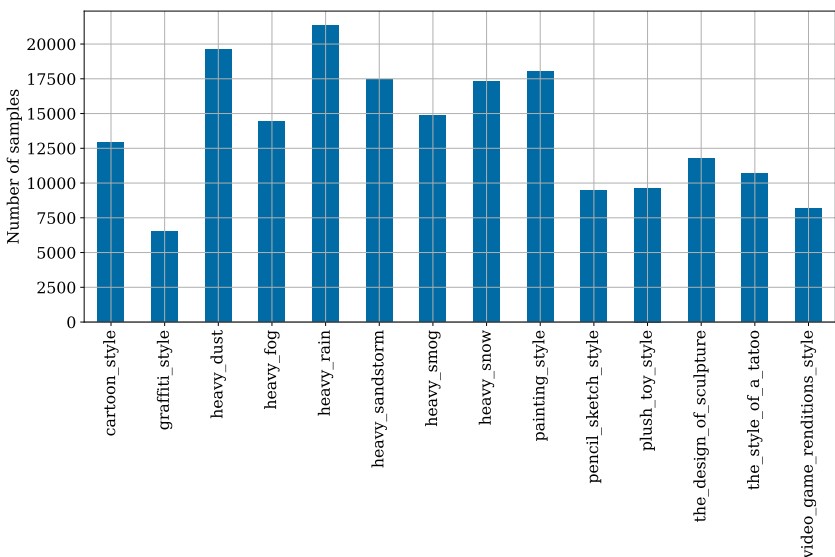

Figure 10: Dataset statistics.

Table 1: **Benchmarked Models.** We present an overview of models that are evaluated and discussed in the main paper by considering their architecture, supervision (supervised, self-supervised, vision-language, or generative), and dataset.

| Model | Architecture | Supervision | Dataset |
|---|---|---|---|
| ResNet | CNN-(18,34,50,101,152) | Classification | IN-1k |
| ViT | ViT-B/16 | Classification | IN-22k/IN-1k |
| DeiT | ViT-B/16 | Classification | IN-1k |
| DeiT-3 | ViT-(S,M,B,L,H)/16 | Classification | IN-1k |
| MAE | ViT-B/16 | SSL | IN-1k |
| MoCov3 | ViT-B/16 | SSL | IN-1k |
| DINOv1 | ViT-B/16 | SSL | IN-1k |
| DINOv2 | ViT-B/16 | SSL | LVD-142M |
| CLIP | ViT-B/16 | VLM | WIT-400M |
| Diff-Class | DiT | Generation | IN-1k |

The accuracies for the diffusion classifier are depicted in Fig. 12. Similar to the discussion in the paper, the results showcase that the generative classifier is less robust than a supervised classifier. We use the DiT-based diffusion classifier trained on ImageNet-1k using the available framework [10] and the default hyper-parameters with a resolution of 256. Due to high computational costs, we compute the results for 25 classes, three scales, for the snow and cartoon style shift, and for at most 10 seeds per class, scale, and shift.

## A.4  Implementation Details

In this section, we provide more implementation details about the dataset generation process.

### A.4.1  Implementation Details for Image Generation

We use the standard diffusers [15] pipeline for Stable Diffusion 2.0, the DDIM sampler with 100 steps and a guidance scale of 7.5, seeds ranging from 1 to 50.

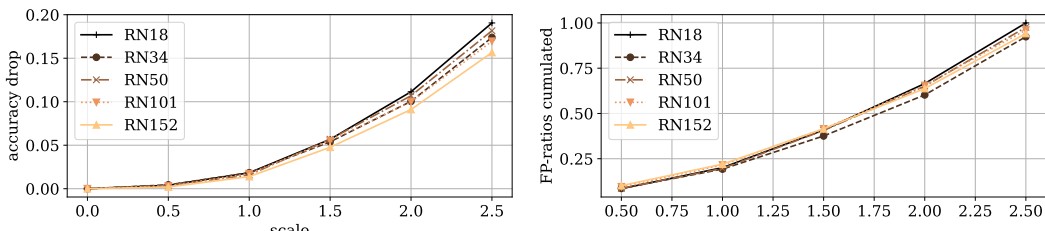

Figure 11: **Robustness evaluation for ResNet model family.** We vary the model size for a set of ResNet models.

Table 2: **Accuracy evaluations.** Accuracies and accuracy drops of all evaluated classifiers.

| | Shift Scale | | | | | | | | | | | |
|---|---|---|---|---|---|---|---|---|---|---|---|---|
| | Accuracy | | | | | | | Accuracy Drop | | | | |
| model | 0 | 0.5 | 1 | 1.5 | 2 | 2.5 | avg | 1 | 1.5 | 2 | 2.5 | avg |
| clip_resnet50 | 0.81 | 0.81 | 0.8 | 0.78 | 0.74 | 0.67 | 0.77 | 0.01 | 0.03 | 0.07 | 0.14 | 0.04 |
| clip_resnet101 | 0.86 | 0.86 | 0.85 | 0.83 | 0.81 | 0.74 | 0.82 | 0.01 | 0.03 | 0.06 | 0.12 | 0.04 |
| clip_vit_base_patch16_224 | 0.87 | 0.88 | 0.88 | 0.87 | 0.86 | 0.81 | 0.86 | -0 | 0.01 | 0.02 | 0.06 | 0.02 |
| clip_vit_base_patch32_224 | 0.87 | 0.87 | 0.86 | 0.85 | 0.83 | 0.77 | 0.84 | 0.01 | 0.02 | 0.04 | 0.1 | 0.03 |
| clip_vit_large_patch14_224 | 0.87 | 0.87 | 0.87 | 0.86 | 0.85 | 0.82 | 0.86 | -0 | 0.01 | 0.02 | 0.05 | 0.01 |
| clip_vit_large_patch14_336 | 0.88 | 0.88 | 0.88 | 0.87 | 0.86 | 0.83 | 0.87 | 0 | 0.01 | 0.02 | 0.05 | 0.01 |
| convnext_tiny.fb_in1k | 0.92 | 0.92 | 0.91 | 0.88 | 0.84 | 0.77 | 0.87 | 0.01 | 0.04 | 0.08 | 0.15 | 0.05 |
| convnext_small.fb_in1k | 0.92 | 0.93 | 0.92 | 0.89 | 0.86 | 0.8 | 0.89 | 0.01 | 0.03 | 0.07 | 0.13 | 0.04 |
| convnext_base.fb_in1k | 0.93 | 0.93 | 0.92 | 0.89 | 0.85 | 0.79 | 0.89 | 0.01 | 0.03 | 0.07 | 0.13 | 0.04 |
| convnext_large.fb_in1k | 0.93 | 0.92 | 0.92 | 0.89 | 0.86 | 0.8 | 0.89 | 0.01 | 0.04 | 0.07 | 0.12 | 0.04 |
| convnextv2_base.fcmae_ft_in1k | 0.93 | 0.93 | 0.92 | 0.9 | 0.87 | 0.82 | 0.9 | 0.01 | 0.04 | 0.07 | 0.12 | 0.04 |
| convnextv2_large.fcmae_ft_in1k | 0.94 | 0.93 | 0.93 | 0.91 | 0.88 | 0.84 | 0.91 | 0.01 | 0.03 | 0.05 | 0.1 | 0.03 |
| convnextv2_huge.fcmae_ft_in1k | 0.94 | 0.93 | 0.93 | 0.91 | 0.89 | 0.84 | 0.91 | 0.01 | 0.03 | 0.05 | 0.09 | 0.03 |
| deit3_small_patch16_224.fb_in1k | 0.92 | 0.92 | 0.91 | 0.88 | 0.84 | 0.77 | 0.87 | 0.01 | 0.04 | 0.08 | 0.15 | 0.05 |
| deit3_base_patch16_224.fb_in1k | 0.91 | 0.91 | 0.9 | 0.88 | 0.84 | 0.79 | 0.87 | 0.01 | 0.03 | 0.07 | 0.12 | 0.04 |
| deit3_medium_patch16_224.fb_in1k | 0.92 | 0.92 | 0.91 | 0.88 | 0.84 | 0.78 | 0.88 | 0.01 | 0.04 | 0.08 | 0.14 | 0.05 |
| deit3_large_patch16_224.fb_in1k | 0.91 | 0.91 | 0.9 | 0.88 | 0.85 | 0.8 | 0.88 | 0.01 | 0.03 | 0.06 | 0.12 | 0.04 |
| deit3_huge_patch14_224.fb_in1k | 0.92 | 0.92 | 0.91 | 0.89 | 0.86 | 0.81 | 0.89 | 0.01 | 0.03 | 0.06 | 0.11 | 0.04 |
| deit_base_patch16_224.fb_in1k | 0.9 | 0.9 | 0.89 | 0.87 | 0.83 | 0.76 | 0.86 | 0.01 | 0.04 | 0.08 | 0.15 | 0.05 |
| dino_vit_base_patch16 | 0.9 | 0.9 | 0.89 | 0.85 | 0.8 | 0.71 | 0.84 | 0.01 | 0.05 | 0.1 | 0.19 | 0.06 |
| dinov2_vit_small_patch14 | 0.92 | 0.92 | 0.91 | 0.89 | 0.86 | 0.81 | 0.89 | 0.01 | 0.03 | 0.06 | 0.11 | 0.04 |
| dinov2_vit_small_patch14_reg | 0.93 | 0.93 | 0.92 | 0.9 | 0.87 | 0.81 | 0.89 | 0.01 | 0.03 | 0.06 | 0.11 | 0.04 |
| dinov2_vit_base_patch14 | 0.91 | 0.91 | 0.91 | 0.89 | 0.87 | 0.82 | 0.89 | 0 | 0.02 | 0.04 | 0.09 | 0.02 |
| dinov2_vit_base_patch14_reg | 0.92 | 0.92 | 0.92 | 0.9 | 0.88 | 0.84 | 0.9 | 0 | 0.02 | 0.04 | 0.08 | 0.02 |
| dinov2_vit_large_patch14 | 0.92 | 0.92 | 0.92 | 0.91 | 0.89 | 0.86 | 0.9 | 0 | 0.01 | 0.03 | 0.06 | 0.02 |
| dinov2_vit_large_patch14_reg | 0.92 | 0.92 | 0.91 | 0.91 | 0.89 | 0.86 | 0.9 | 0 | 0.01 | 0.03 | 0.06 | 0.02 |
| dinov2_vit_giant_patch14 | 0.91 | 0.91 | 0.91 | 0.9 | 0.88 | 0.84 | 0.89 | 0 | 0.01 | 0.04 | 0.07 | 0.02 |
| dinov2_vit_giant_patch14_reg | 0.92 | 0.92 | 0.91 | 0.9 | 0.88 | 0.85 | 0.9 | 0 | 0.01 | 0.03 | 0.07 | 0.02 |
| mae_vit_base_patch16 | 0.92 | 0.92 | 0.91 | 0.88 | 0.84 | 0.78 | 0.88 | 0.01 | 0.04 | 0.08 | 0.14 | 0.05 |
| mae_vit_huge_patch14 | 0.93 | 0.93 | 0.92 | 0.9 | 0.88 | 0.84 | 0.9 | 0.01 | 0.03 | 0.05 | 0.1 | 0.03 |
| mae_vit_large_patch16 | 0.93 | 0.92 | 0.92 | 0.9 | 0.87 | 0.83 | 0.9 | 0.01 | 0.03 | 0.05 | 0.1 | 0.03 |
| mocov3_vit_base_patch16 | 0.92 | 0.92 | 0.91 | 0.88 | 0.85 | 0.79 | 0.88 | 0.01 | 0.03 | 0.07 | 0.13 | 0.04 |
| resnet18.a1_in1k | 0.9 | 0.9 | 0.88 | 0.85 | 0.8 | 0.72 | 0.84 | 0.02 | 0.05 | 0.1 | 0.19 | 0.06 |
| resnet34.a1_in1k | 0.91 | 0.91 | 0.9 | 0.86 | 0.82 | 0.75 | 0.86 | 0.01 | 0.05 | 0.09 | 0.17 | 0.05 |
| resnet50.a1_in1k | 0.91 | 0.9 | 0.89 | 0.85 | 0.8 | 0.72 | 0.85 | 0.02 | 0.06 | 0.11 | 0.18 | 0.06 |
| resnet101.a1_in1k | 0.9 | 0.9 | 0.88 | 0.85 | 0.8 | 0.73 | 0.84 | 0.02 | 0.05 | 0.1 | 0.17 | 0.06 |
| resnet152.a1_in1k | 0.89 | 0.89 | 0.88 | 0.85 | 0.8 | 0.73 | 0.84 | 0.01 | 0.04 | 0.09 | 0.16 | 0.05 |
| vit_base_patch16_224.augreg_in1k | 0.87 | 0.87 | 0.86 | 0.82 | 0.77 | 0.69 | 0.81 | 0.01 | 0.05 | 0.1 | 0.18 | 0.06 |
| vit_base_patch16_224.augreg_in21k_ft_in1k | 0.9 | 0.9 | 0.89 | 0.86 | 0.82 | 0.75 | 0.85 | 0.01 | 0.04 | 0.08 | 0.15 | 0.05 |
| vit_base_patch16_clip_224.openai_ft_in1k | 0.93 | 0.93 | 0.92 | 0.91 | 0.89 | 0.86 | 0.91 | 0.01 | 0.02 | 0.04 | 0.08 | 0.03 |

### A.4.2 Ablation of Image Generation

We ablate how the number of classes influences the robustness evaluations in Fig. 13. For a more efficient computation, we use the `UniPCMultistepScheduler` sampler with 20 steps [16]. In addition to 100 sliders for 14 shifts, we also publish the sliders for all 1000 ImageNet classes for the shifts snow and cartoon.

### A.4.3 Text-Based Continuous Shift

Following the implementation of Baumann et al. [1], we explore whether continuous shifts can be applied in a naive way and we present some examples in Fig. 14. We achieve reasonable results for some classes (*e.g.*, upper row). However, we observe the following issues arising from this strategy: (1) The semantic structures clearly change, which involves other factors of variation. This does not allow the computation of a failure point along one sliding trajectory. (2) depicted in middle row: For some classes, the naive approach is very unstable, resulting in OOD samples that do not

Table 3: **More results for failure distribution**. We report the ratio of failure points for all models, where the sum of all failure points is normalized for each model.

| model | Shift Scale | | | | | |
| --- | --- | --- | --- | --- | --- | --- |
| | 0 | 0.5 | 1 | 1.5 | 2 | 2.5 |
| clip_resnet50 | 0.28 | 0.06 | 0.07 | 0.11 | 0.17 | 0.31 |
| clip_resnet101 | 0.23 | 0.06 | 0.08 | 0.12 | 0.18 | 0.33 |
| clip_vit_base_patch16_224 | 0.27 | 0.04 | 0.06 | 0.1 | 0.16 | 0.37 |
| clip_vit_base_patch32_224 | 0.24 | 0.05 | 0.06 | 0.1 | 0.18 | 0.36 |
| clip_vit_large_patch14_224 | 0.29 | 0.05 | 0.07 | 0.13 | 0.15 | 0.31 |
| clip_vit_large_patch14_336 | 0.27 | 0.05 | 0.07 | 0.11 | 0.17 | 0.33 |
| convnext_tiny.fb_in1k | 0.14 | 0.04 | 0.06 | 0.13 | 0.22 | 0.42 |
| convnext_small.fb_in1k | 0.15 | 0.03 | 0.08 | 0.18 | 0.22 | 0.34 |
| convnext_base.fb_in1k | 0.15 | 0.04 | 0.06 | 0.14 | 0.23 | 0.36 |
| convnext_large.fb_in1k | 0.16 | 0.04 | 0.08 | 0.19 | 0.21 | 0.33 |
| convnextv2_base.fcmae_ft_in1k | 0.14 | 0.04 | 0.06 | 0.13 | 0.23 | 0.4 |
| convnextv2_large.fcmae_ft_in1k | 0.15 | 0.04 | 0.07 | 0.16 | 0.2 | 0.38 |
| convnextv2_huge.fcmae_ft_in1k | 0.14 | 0.04 | 0.06 | 0.14 | 0.21 | 0.41 |
| deit3_small_patch16_224.fb_in1k | 0.16 | 0.05 | 0.07 | 0.15 | 0.22 | 0.36 |
| deit3_medium_patch16_224.fb_in1k | 0.15 | 0.04 | 0.06 | 0.15 | 0.23 | 0.36 |
| deit3_base_patch16_224.fb_in1k | 0.17 | 0.03 | 0.06 | 0.14 | 0.23 | 0.36 |
| deit3_large_patch16_224.fb_in1k | 0.18 | 0.04 | 0.07 | 0.14 | 0.21 | 0.36 |
| deit3_huge_patch14_224.fb_in1k | 0.17 | 0.04 | 0.06 | 0.14 | 0.21 | 0.37 |
| deit_base_patch16_224.fb_in1k | 0.18 | 0.04 | 0.08 | 0.16 | 0.2 | 0.33 |
| dino_vit_base_patch16 | 0.16 | 0.04 | 0.07 | 0.17 | 0.22 | 0.35 |
| dinov2_vit_small_patch14 | 0.15 | 0.04 | 0.07 | 0.14 | 0.24 | 0.35 |
| dinov2_vit_small_patch14_reg | 0.15 | 0.04 | 0.07 | 0.16 | 0.21 | 0.36 |
| dinov2_vit_base_patch14 | 0.19 | 0.05 | 0.07 | 0.12 | 0.19 | 0.38 |
| dinov2_vit_base_patch14_reg | 0.2 | 0.06 | 0.07 | 0.13 | 0.2 | 0.35 |
| dinov2_vit_large_patch14 | 0.2 | 0.06 | 0.07 | 0.12 | 0.19 | 0.36 |
| dinov2_vit_large_patch14_reg | 0.23 | 0.05 | 0.07 | 0.1 | 0.19 | 0.36 |
| dinov2_vit_giant_patch14 | 0.21 | 0.04 | 0.07 | 0.1 | 0.2 | 0.37 |
| dinov2_vit_giant_patch14_reg | 0.22 | 0.05 | 0.08 | 0.12 | 0.19 | 0.35 |
| mae_vit_base_patch16 | 0.15 | 0.04 | 0.07 | 0.14 | 0.22 | 0.38 |
| mae_vit_large_patch16 | 0.18 | 0.04 | 0.08 | 0.13 | 0.2 | 0.37 |
| mae_vit_huge_patch14 | 0.15 | 0.04 | 0.06 | 0.15 | 0.21 | 0.39 |
| mocov3_vit_base_patch16 | 0.16 | 0.04 | 0.07 | 0.14 | 0.22 | 0.38 |
| resnet18.a1_in1k | 0.16 | 0.05 | 0.07 | 0.18 | 0.2 | 0.34 |
| resnet34.a1_in1k | 0.16 | 0.05 | 0.08 | 0.16 | 0.23 | 0.33 |
| resnet50.a1_in1k | 0.16 | 0.05 | 0.07 | 0.16 | 0.24 | 0.32 |
| resnet101.a1_in1k | 0.18 | 0.05 | 0.08 | 0.16 | 0.22 | 0.32 |
| resnet152.a1_in1k | 0.18 | 0.05 | 0.08 | 0.19 | 0.2 | 0.29 |
| vit_base_patch16_224.augreg_in1k | 0.21 | 0.04 | 0.07 | 0.17 | 0.2 | 0.3 |
| vit_base_patch16_224.augreg_in21k_ft_in1k | 0.16 | 0.04 | 0.08 | 0.16 | 0.21 | 0.36 |
| vit_base_patch16_clip_224.openai_ft_in1k | 0.16 | 0.05 | 0.07 | 0.14 | 0.18 | 0.39 |

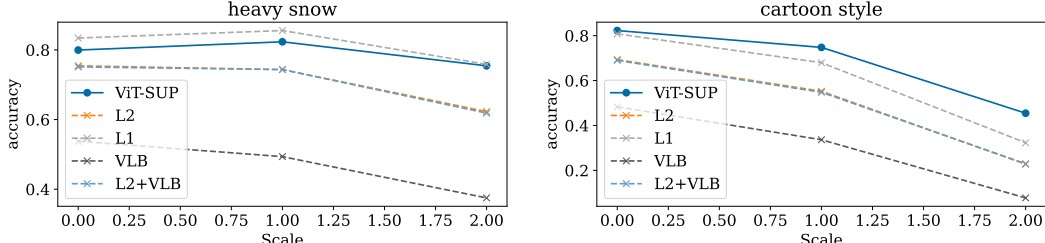

Figure 12: **Results for diffusion-classifier DiT.** We report the classification for three scales and the four configurations for computing the classes, as proposed in Li et al. [10].

represent realistic images. We did not reach significantly better results when applying a delayed sampling technique for the delta embedding. (3) depicted in the bottom row: Applying the delta in text-embedding space does not always result in a consistent increase of the considered shift.

### A.4.4 Implementation Details for Benchmarking

We provide the code for training the LoRA adapters and for performing the sliding. For benchmarking all vision models, we integrate our new benchmark and additional models in the easyrobust [11] framework. We provide all classification results for all images of the dataset together with the code and the data in the supplementary material.

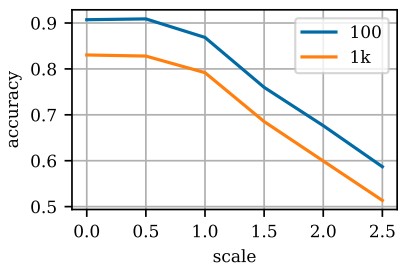

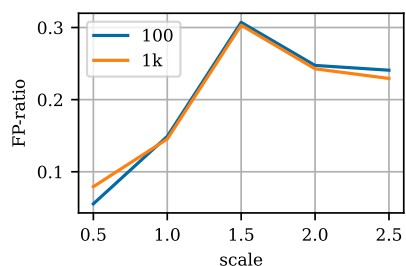

(a) Accuracy over various scales.

(b) Failure point distribution (normalized over the sum of failure points).

Figure 13: **Ablation of the number of ImageNet classes.**. We compare the accuracies and failure points averaged over the selected 100 classes and all 1000 ImageNet classes for two shifts (snow and cartoon style). We report the results with ResNet-50.

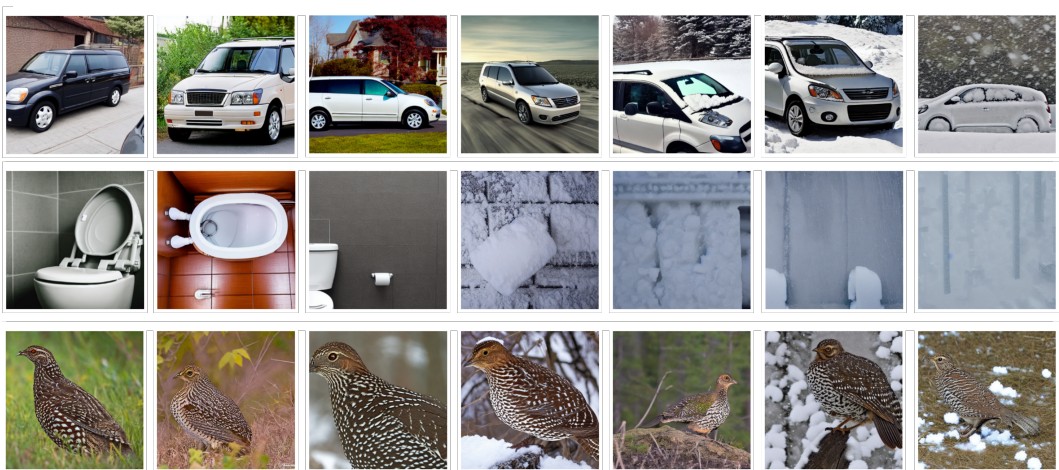

Figure 14: **Examples for text-based continuous shift.** The gradual increase can be successful. However, we observe that it fails for some classes (middle row) and is not consistently increasing (bottom row).

### A.4.5 Details about the Used Compute

We used the internal cluster consisting of NVIDIA A40, A100, and RTX 8000 GPUs for running most of the experiments. Small-scale experiments are conducted on workstations equipped with RTX 3090. Training one LoRA adapter requires 1 to 2 hours (A100 / A40), generating the images for 14 shifts, 100 classes, 50 seeds, and 6 scales, requires 10 to 20 minutes, which, respectively, equaled around 2000 GPU hours and around 7500 GPU hours for the published benchmark in total. Benchmarking all models of *easyrobust* required around 1000 GPU hours. The experiments to perform classification using the diffusion-classifier require around 4000 GPU hours.

### A.5 Labeling

We refer to Sec. 4 for the explanation of the filtering. In this section, we provide more details about the labeling strategy and its statistics.

### A.5.1 Discussion and Statistics of Labeling Strategy

For the pre-filtering strategy (ii) and for the selection of easy samples (iii), we compute text-alignment using CLIP score and we remove all samples that have a CLIP similarity $s_{\text{CLIP-text-alignment}} > 24$, which approximately includes 90% of all ImageNet validation images [14]. We use the implementation in

*torchmetrics* with VIT-B/16. For the correct classification in (ii) and (iii), we consider the following classifiers: ResNet-50 [9], ViT-B/16 [4], DeiT-B/16 [13]. For DINOv2, we apply DINOv2-R-ViT-L [2, 12] with a linear head. After removing the easy samples in step (iii), 2.7k images remain for labeling. We use the VIA annotation tool [5, 6] to create the annotations. Each image is labeled by two humans. In total, 14 graduate students are involved in the labeling process. For all participants, we ensure sufficient motivation and they receive detailed instructions on how to perform the labeling (the full set of instructions is provided in Fig. 18). We provide the filtering statistics in Tab. 4. An example screenshot of the labeling tool is visualized in Fig. 15.

Table 4: **Statistics of filtering process.** We report the number of samples after various filtering stages. The stages are numbered according to the description in the main paper.

| Scale | Stage (i) | Stage (ii) | Stage (iii) | Stage (iv) |
|---|---|---|---|---|
| 0 | 4000 | 2966 | 2966 | 2966 |
| 0.5 | 4000 | 2966 | 2929 | 2955 |
| 1 | 4000 | 2966 | 2813 | 2906 |
| 1.5 | 4000 | 2966 | 2479 | 2740 |
| 2 | 4000 | 2966 | 2143 | 2498 |
| 2.5 | 4000 | 2966 | 1729 | 2110 |

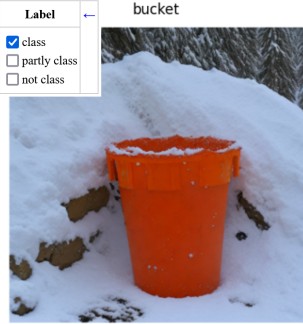

Figure 15: Screenshot of labeling tool.

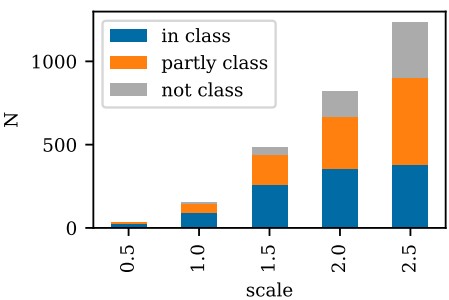

(a) For the human labeling dataset.

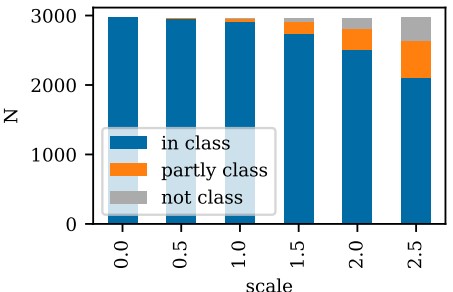

(b) For the complete filtering dataset.

Figure 16: **Statistics of labeling dataset.** We report the number of in-class, partially in-class, and out-of-class samples.

### A.5.2 Labeling Dataset

We provide the images for labeling in the provided URL as well. There, we include all images and metadata that allow inferring the class of each image and the tag, whether it is labeled automatically or by a human. The statistics of the labeling dataset are shown in Fig. 16.

### A.6 Weaknesses

In this section, we discuss the weaknesses of our method.

#### A.6.1 Weaknesses of Filtering Strategy

Applying an automated filtering strategy comes with two challenges:
(1) While we showcase that our filtering strategy achieves a high accuracy on the labeled dataset, the application of surrogate models based on CLIP or DINOv2 sometimes removes samples from the benchmark that are actually in-class samples. However, our applied filtering strategies fail to recognize this, which biases the benchmark.
(2) Our filtering algorithm does not remove all out-of-class samples. This needs to be considered carefully when analyzing the accuracy drop for one specific model and style shift. We are, however, interested in comparing the accuracy drops for various classifiers, which are equally affected by out-of-class samples.

#### A.6.2 Weaknesses of Benchmark

While we perform the analysis on 14 diverse shifts, including not only natural variations but also style shifts, this list does not completely represent all real-world nuisance shifts. Therefore, the robustness estimate is only an approximation of the robustness in arbitrary shifts. However, our framework allows for the addition of arbitrary shifts, and we motivate the community to provide more shifts. In addition, we encourage to compute the robustness with respect to individual nuisance shifts.

### A.7 OOD-CV Details

The Out-of-Distribution Benchmark for Robustness (OOD-CV) dataset includes real-world OOD examples of 10 object categories varying in terms of 5 nuisance factors: *pose*, *shape*, *context*, *texture*, and *weather*.

**Generation of images for synthetic OOD-CV**   We generate the images for the synthetic OOD-CV dataset using a larger number of noise steps (85%) and more scale (between 0 and 3) since the classes occur more often in the dataset for training CLIP and Stable Diffusion. We use SD2.0 and not the dataset interfaces provided by Vendrow et al. [14] since the class differences are less subtle and the samples of OOD-CV originate from two different datasets.

**Training subset**   The OOD-CV benchmark provides a training subset of 8627 images. We train different state-of-the-art classifiers (i.e., ResNet-50 [9], ViT-B/16 [4], and DINO-v2-ViT [12]) for classification. We finetune each baseline during 50 epochs with an early stopping set to 5 epochs. In order to make baselines more robust, we apply standard data augmentation such as scale, rotation, and flipping during training. The training subset is composed of images originating from different datasets, notably ImageNet [3] and Pascal-VOC [7]. It is important to notice that the distribution of these two subsets is slightly different, with a higher data quality for the ImageNet subset and a lower quality for the latter subset (more noise, smaller objects, different image sizes). We visualize a few examples of the training data in Fig. 17.

**Test subset annotations**   In the test subset provided in the benchmark dataset, only the coarse individual nuisance factors (*e.g.*, *weather*, *texture*) are provided. In our setup, we are interested in studying more fine-grained nuisance shifts, notably *rain*, *snow*, or *fog*. Hence, we had to assign some fine-grained annotation to all images containing *weather* nuisance shifts.   Hence, we assign a fine-grained annotation by computing the CLIP sim-ilarity to the following texts: "a picture of a {class} in {shift}", where class is the ground truth class and shift the nuisance shift candidate *rain*, *snow*, or *fog* and "a picture of a {class} without snow nor fog nor rain". By applying a softmax on the similarity scores with the previous texts, we can assign the fine-grained nuisance shift *rain*, *snow*, *fog* or *unknown* for each image. We show more statistics in Tab. 5. By checking the results visually,

we observe that all fine-grained nuisance shifts align with human perception and have a tendency towards classifying samples as *unknown* as soon as there is a small doubt. Note that by applying the same strategies to our generated data, we obtain an accuracy close to $100\%$. Please note that our generated data has been automatically filtered using a similar approach as described previously and verified manually for the four studied nuisance shifts in order to make sure that the comparison with the OOD-CV benchmark was consistent. The filtered data can be found in the GoogleDrive previously mentioned.

Table 5: **OOD-CV Statistics.** We report the number of images and accuracies for the weather subset.

| Shift | #images | Accuracy |
|---|---|---|
| Snow | 273 | 70.3 |
| Fog | 24 | 62.5 |
| Rain | 74 | 66.2 |
| Unknown | 129 | 66.7 |
| Total | 500 | 68.4 |

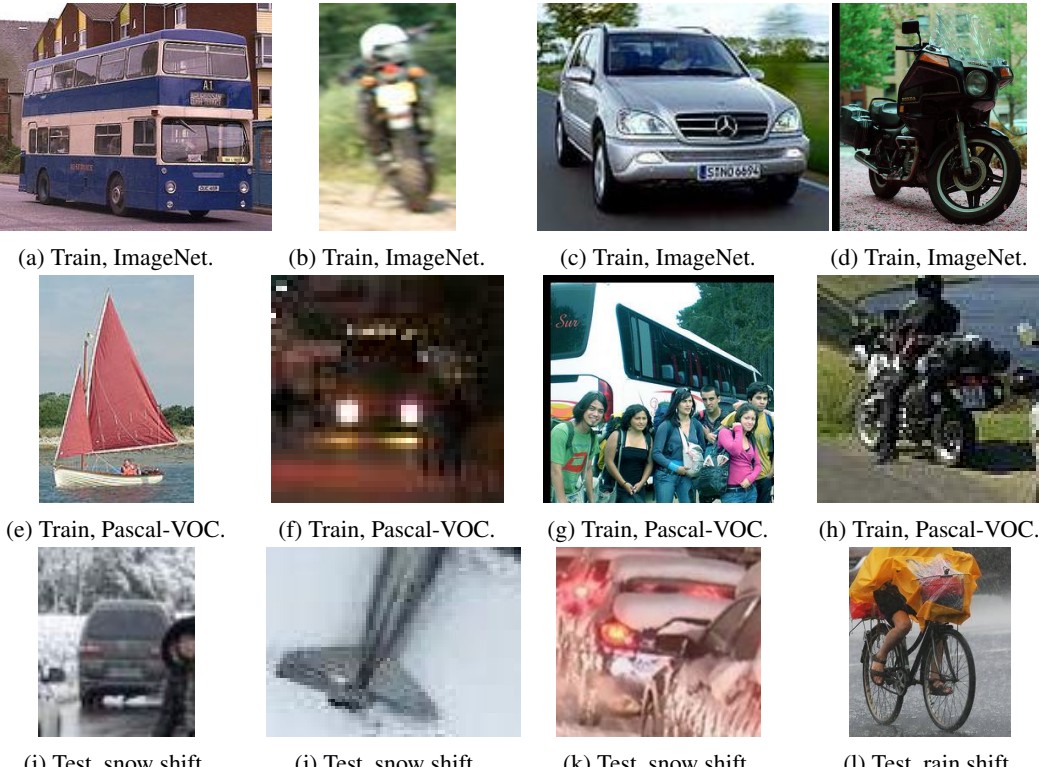

(a) Train, ImageNet.    (b) Train, ImageNet.    (c) Train, ImageNet.    (d) Train, ImageNet.

(e) Train, Pascal-VOC.    (f) Train, Pascal-VOC.    (g) Train, Pascal-VOC.    (h) Train, Pascal-VOC.

(i) Test, snow shift.    (j) Test, snow shift.    (k) Test, snow shift.    (l) Test, rain shift.

Figure 17: **OOD-CV example images.** We illustrate a set of example images from the training and the testing dataset of OOD-CV: (a-h) example from the training set, from ImageNet or Pascal-VOC. (i-l) Some examples for weather nuisance shifts. In the training set, we observe that images from the Pascal-VOC subset are usually of lower quality (*e.g.*, cropping, occlusion, resolution) compared to the ImageNet subset. In the test set, we see that that not fully disentangled (*e.g.*, (j) is only partially visible, (k) is partially occluded).

# Labeling task for out-of-class detection

**Motivation**: For benchmarking a classifier with synthetic images, we need to ensure that the generated images still correspond to the correct classes. To evaluate automatic filtering pipelines, we create a dataset with human labels. The dataset includes generated images with various levels of snow or cartoon style.

**Task:**
The goal is to detect images that do not belong to the corresponding ImageNet class (given as title).

Given an image, your task is to select one of three labels:
- ***class*:**
  - You can clearly recognize the class.
- ***partly class*:**
  - Given the class label, the class seems to correspond to the image.
  - You can recognize parts of the class but you are not very sure whether this is actually the class
  - You clearly see some characteristics of the class but it does not include all the important features.
- ***not class*:**
  - The considered image is clearly not the considered class.

The goal is to check whether the objects in the image correspond to a class or not. The goal is not to check whether the samples look realistic.

Every class starts with one realistic example image, taken from ImageNet. This image needs to be labeled as well. Since the example is just one illustrative example, not depicting the diversity of the class, it is recommended to use Google picture search to get an intuition of how the object looks in case one is not familiar with the class.
Some of the consecutive class samples will be similar. They are generated with the same seed but with varying snow or cartoon levels.

Some examples for class, partly class, and not class:

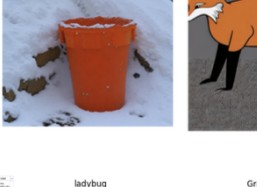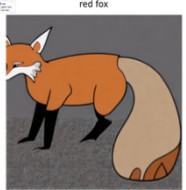

1) ***class*:** This animal can be clearly described as a fox at first glance. Also, the bucket can be easily recognized.

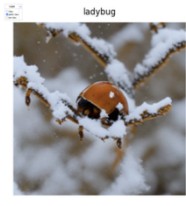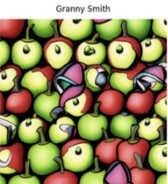

2) ***partly class*:** The shape and size seems to fit a ladybug. However, the black dots are missing. The other picture might be a cartoon-like illustration of apples. However, this can be argued. It is not clear.

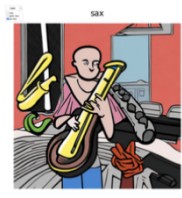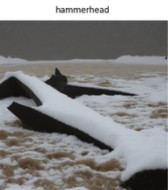

3) ***not class*:** First example: This is supposed to be a sax but it is clearly not recognizable as a sax. Second example: There is not a single characteristic that resembles a hammerhead. It is very clearly not the class.

Figure 18: **Set of instructions for labeling.** We provided the instructions provided to the human annotators to perform the labeling of the out-of-class filtering dataset.

# B  Datasheet

In the following, we answer the questions as proposed in Gebru et al. [8].

## B.1  Motivation

**For what purpose was the dataset created?** Was there a specific task in mind? Was there a specific gap that needed to be filled? Please provide a description.

The dataset was created to evaluate the robustness of state-of-the-art models to specific continuous nuisance shifts. Current approaches are not scalable and often include only a small variety of nuisance shifts, which are not always relevant in the real world. More importantly, current benchmark datasets define binary nuisance shifts by considering the existence or absence of that shift, which may contradict their continuous realization in real-world scenarios.

**Who created the dataset (e.g., which team, research group) and on behalf of which entity (e.g., company, institution, organization)?**

Until the acceptance of the paper, the specific details about the research group, their affiliations, and the entities they represent will remain anonymous.

**Who funded the creation of the dataset?** If there is an associated grant, please provide the name of the grantor and the grant name and number.

Until the acceptance of the paper, the specific details about funding will remain anonymous.

## B.2  Composition

**What do the instances that comprise the dataset represent (e.g., documents, photos, people, countries)?**

The dataset consists of synthetic images that were generated using Stable Diffusion.

**How many instances are there in total (of each type, if appropriate)?**

The dataset contains $192,168$ images in total, with $32,028$ for each of the six scales with 14 shifts. Each shift has at least $5,000$ images and 100 classes.

**Does the dataset contain all possible instances or is it a sample (not necessarily random) of instances from a larger set?** If the dataset is a sample, then what is the larger set? Is the sample representative of the larger set (e.g., geographic coverage)? If so, please describe how this representativeness was validated/verified. If it is not representative of the larger set, please describe why not (e.g., to cover a more diverse range of instances because instances were withheld or unavailable).

The dataset contains the subset of images that were filtered using the selected filtering strategy. Originally, $420,000$ images were generated.

**What data does each instance consist of?  "Raw" data (e.g., unprocessed text or images) or features?** In either case, please provide a description.

"Raw" synthetically generated data as described in the paper.

**Is there a label or target associated with each instance?** If so, please provide a description.

Yes, each image belongs to an ImageNet class and has a shift scale assigned to it.

**Is any information missing from individual instances?** If so, please provide a description, explaining why this information is missing (e.g., because it was unavailable). This does not include intentionally removed information, but might include, e.g., redacted text.

No, for each instance, we give the class label, the scale of the shift, and the parameters used for generating this image. However, the class label might be erroneous in rare cases where the generated image corresponds to an out-of-class sample.

**Are relationships between individual instances made explicit (e.g., users with their tweets, songs with their lyrics, nodes with edges)?** If so, please describe how these relationships are made explicit.

Yes, the relationships in terms of class, random seed for generation, shift, and scale of shift are provided in the dataset.

**Are there recommended data splits (e.g., training, development/validation, testing)?** If so, please provide a description of these splits, explaining the rationale behind them.

We offer a benchmark dataset specifically intended for testing the robustness of classifiers. Therefore, we recommend utilizing the entire dataset provided as the test dataset.

**Are there any errors, sources of noise, or redundancies in the dataset?** If so, please provide a description.

We provided a dataset of generated images. While we apply a filtering strategy to reduce the number of out-of-class and unrealistic samples, we cannot guarantee that all images of the dataset represent a realistic and visually appealing realization of the considered class. We provide a statistical estimate of the number of failure samples in the paper. The data might also include the redundancies that underlie the image generation process of Stable Diffusion.

**Is the dataset self-contained, or does it link to or otherwise rely on external resources (e.g., websites, tweets, other datasets)?** If it links to or relies on external resources, a) are there guarantees that they will exist, and remain constant, over time; b) are there official archival versions of the complete dataset (i.e., including the external resources as they existed at the time the dataset was created); c) are there any restrictions (e.g., licenses, fees) associated with the use of these external resources?

The dataset is fully self-contained.

**Does the dataset contain data that might be considered confidential (e.g., data that is protected by legal privilege or by doctor–patient confidentiality, data that includes the content of individuals' non-public communications)?** If so, please provide a description.

No.

**Does the dataset contain data that, if viewed directly, might be offensive, insulting, threatening, or might otherwise cause anxiety?** If so, please describe why.

There is a small chance that our synthetically generated data can generate offensive images. However, we did not encounter any such sample during our extensive manual annotations.

**Does the dataset relate to people? If not, you may skip the remaining questions in this section.**

No.

**Does the dataset identify any subpopulations (e.g., by age, gender)?** If so, please describe how these subpopulations are identified and provide a description of their respective distributions within the dataset.

N/A.

**Is it possible to identify individuals (i.e., one or more natural persons), either directly or indirectly (i.e., in combination with other data) from the dataset?** If so, please describe how.

N/A.

**Does the dataset contain data on individuals' protected characteristics (e.g., age, gender, race, religion, sexual orientation)?** If so, please describe this data and how it was obtained.

N/A.

**Does the dataset contain data on individuals' criminal history or other behaviors that would typically be considered sensitive or confidential?** If so, please describe this data and how it was obtained.

N/A.

### B.3  Collection Process

**How was the data associated with each instance acquired? Was the data directly observable (e.g., raw text, movie ratings), reported by subjects (e.g., survey responses), or indirectly inferred/derived from other data (e.g., part-of-speech tags, model-based guesses)?**

N/A.

**What mechanisms or procedures were used to collect the data (e.g., hardware apparatus or sensor, manual human curation, software program, software API)? How were these mechanisms or procedures validated?**

We used Stable Diffusion 2.0 to generate all images. Images were generated using NVIDIA A100 and A40 GPUs.

**If the dataset is a sample from a larger set, what was the sampling strategy (e.g., deterministic, probabilistic with specific sampling probabilities)?**

The dataset was filtered using a combinatorial selection approach using DINOv2-R and a CLIP model.

**Who was involved in the data collection process (e.g., students, crowdworkers, contractors) and how were they compensated (e.g., how much were crowdworkers paid)?**

The authors of the paper. They were not additionally paid for the dataset collection process.

**Over what timeframe was the data collected? Does this timeframe match the creation timeframe of the data associated with the instances (e.g., recent crawl of old news articles)?** If not, please describe the timeframe in which the data associated with the instances was created.

The images were generated and processed over a timeframe of four weeks.

**Were any ethical review processes conducted (e.g., by an institutional review board)?** If so, please provide a description of these review processes, including the outcomes, as well as a link or other access point to any supporting documentation.

No ethical concerns.

### B.4  Preprocessing/cleaning/labeling

**Was any preprocessing/cleaning/labeling of the data done (e.g., discretization or bucketing, tokenization, part-of-speech tagging, SIFT feature extraction, removal of instances, processing of missing values)?** If so, please provide a description. If not, you may skip the remaining questions in this section.

Yes, cleaning of the generated data was conducted. The generated images underwent filtering to reduce the number of out-of-class samples using the proposed filtering mechanisms. Instances that did not meet these criteria were removed from the dataset. For a detailed description of the filtering process, please refer to the corresponding section in the paper.

**Was the "raw" data saved in addition to the preprocessed/cleaned/labeled data (e.g., to support unanticipated future uses)?** If so, please provide a link or other access point to the "raw" data.

The generated images remain in their original, unprocessed state and can be considered as "raw" data. However, we have not provided all the data that was filtered out during filtering.

**Is the software used to preprocess/clean/label the instances available?** If so, please provide a link or other access point.

Generating the images was performed using commonly available Python libraries. For annotating a subset of the dataset for filtering purposes, we have used the VIA annotation tool [5, 6].

## B.5 Uses

**Has the dataset been used for any tasks already?** If so, please provide a description.

In our work, we demonstrate how this approach yields valuable insights into the robustness of state-of-the-art models, particularly in the context of classification tasks.

**Is there a repository that links to any or all papers or systems that use the dataset?** If so, please provide a link or other access point.

Yes, used and benchmarked systems are cited in the paper. In addition, will add the relevant works in the repository that will provide the code.

**What (other) tasks could the dataset be used for?**

Our work showcases the capability of our dataset to enhance control over data generation, which is particularly evident through continuous shifts. However, its applicability extends beyond this demonstration. The dataset can be effectively utilized in various generation tasks that necessitate continuous parameter control. While we showcased its efficacy in providing insights for models tackling classification tasks, it can seamlessly extend to evaluate the robustness of state-of-the-art methods across diverse tasks such as segmentation, domain adaptation, and many others. This is possible by combining our approach with other modes of conditioning Stable Diffusion.

**Is there anything about the composition of the dataset or the way it was collected and cleaned that might impact future uses? For example, is there anything that might cause the dataset to be used inappropriately or misinterpreted (e.g., accidentally incorporating biases, reinforcing stereotypes)?**

Our dataset was synthesized using a generative model. It, therefore, likely inherits any biases for its generator. Similarly, filtering is performed by a large pre-trained model, which can indirectly also contribute to biases.

**Are there tasks for which the dataset should not be used?** If so, please provide a description.

No, there are no tasks for which the dataset should not be used. Our dataset aims to enhance model robustness and provide deeper insights during model evaluation. Therefore, we see no reason to restrict its usage.

## B.6 Distribution

**Will the dataset be distributed to third parties outside of the entity (e.g., company, institution, organization) on behalf of which the dataset was created?** If so, please provide a description.

Yes, the dataset will be publicly available on the internet.

**How will the dataset be distributed (e.g., tarball on website, API, GitHub)? Does the dataset have a digital object identifier (DOI)?**

In the future, we will distribute the dataset as a tarball on our servers.

**When will the dataset be distributed?**

The dataset will be distributed upon acceptance of the manuscript. Therefore, if accepted, distribution will commence from the end of September 2024. It is now available under the provided anonymized link.

**Will the dataset be distributed under a copyright or other intellectual property (IP) license, and/or under applicable terms of use (ToU)?** If so, please describe this license and/or ToU, and provide a link or other access point to, or otherwise reproduce, any relevant licensing terms or ToU.

CC-BY-NC.

**Have any third parties imposed IP-based or other restrictions on the data associated with the instances?** If so, please describe these restrictions, and provide a link or other access point to, or otherwise reproduce, any relevant licensing terms.

No, there are no IP-based or other restrictions on the data associated with the instances imposed by third parties.

**Do any export controls or other regulatory restrictions apply to the dataset or to individual instances?** If so, please describe these restrictions, and provide a link or other access point to, or otherwise reproduce, any supporting documentation.

We are not aware of any export controls or other regulatory restrictions that apply to the dataset or to individual instances.

### B.7  Maintenance

**Who is supporting/hosting/maintaining the dataset?**

The dataset is supported by the authors and their associated research groups. The dataset is hosted on our own servers.

**How can the owner/curator/manager of the dataset be contacted (e.g., email address)?**

The authors of this dataset will be reachable at their e-mail addresses: [undisclosed]. In addition, we will add a contact form, which will be made available on the website.

**Is there an erratum?** If so, please provide a link or other access point.

If errors are found, an erratum will be added to the website.

**Will the dataset be updated (e.g., to correct labeling errors, add new instances, delete instances)?** If so, please describe how often, when, and how updates will be provided.

Yes, updates will be communicated via the website. The dataset will be versioned.

**If the dataset relates to people, are there applicable limits on the retention of the data associated with the instances (e.g., were individuals in question told that their data would be retained for a specific period of time and then deleted)?** If so, please describe these limits and explain how they will be enforced.

Our dataset does not relate to people.

**Will older versions of the dataset continue to be supported/hosted/maintained?** If so, please describe how.

No, older versions of the dataset will not be supported if the dataset is updated. We do not plan to extend or update the dataset. Any updates will be made solely to correct any hypothetical errors that may be discovered.

**If others want to extend/augment/build on/contribute to the dataset, is there a mechanism for them to do so?** If so, please provide a description. Will these contributions be made publicly available?

Yes, we provide all the necessary tools and explanations to enable users to build continuous shifts for their own specific applications. Our dataset serves as a foundation to illustrate how it can be used to evaluate current state-of-the-art methods. However, we are happy to centralize and showcase all related work on our GitHub page that benefits from our method of generating data.

### B.8  Author Statement of Responsibility

The authors confirm all responsibility in case of violation of rights and confirm the license associated with the dataset and its images.