# OpenReview forum: "Benchmarking Vision Models Under Generative Continuous Nuisance Shifts"
_NeurIPS.cc/2024/Datasets_and_Benchmarks_Track — Submitted to NeurIPS 2024 Track Datasets and Benchmarks_

### Official Review · Reviewer_Hf4n · 2024-07-24
**Lack of evaluation on the realism of the proposed benchmarks**

**Rating:** 3
**Confidence:** 4

**Review:**

### Quality
Although the paper attempts to introduce the proposed benchmark as accurately as possible, it does not describe concrete formal algorithms or protocols, nor does it provide code, making it difficult to verify reproducibility and factuality.

### Clarity
The motivation to reproduce real-world distribution shifts with the benchmark is clear. On the other hand, this motivation is not clear because the paper does not provide evidence of the extent to which the evaluation by the synthetic data set with the proposed benchmark reproduces real-world distribution shifts or how well it correlates with real-world distribution shifts.

### Originality
There is some originality in the focus on using a synthetic dataset to represent distribution shifts with continuous change. However, there is little evidence to support why continuous change is important, limiting its impact. In particular, it is unclear in what cases the proposed benchmark is effective, as distribution shifts may be caused by non-continuous factors (e.g., style).

### Significance
The paper claims that the proposed benchmark has an advantage over the existing OOD-CV benchmark, which consists of real data in terms of its ability to detect change points by distribution shift intensity. However, the significance of the detected change points is unclear because it is not shown whether the continuity of the intensity of the distribution shift by the generative model is related to the continuity of reality.
In addition, most of the findings in the paper are common to and known from existing studies [a,b,c].
From these, the significance of the paper is limited.

[a] Zhao, Bingchen, et al. "Ood-cv: A benchmark for robustness to out-of-distribution shifts of individual nuisances in natural images." European conference on computer vision. Cham: Springer Nature Switzerland, 2022.

[b] Chen, Shuo, et al. "Benchmarking robustness of adaptation methods on pre-trained vision-language models." Advances in Neural Information Processing Systems 36 (2023).

[c] Zhang, Chenshuang, et al. "ImageNet-D: Benchmarking Neural Network Robustness on Diffusion Synthetic Object." Proceedings of the IEEE/CVF Conference on Computer Vision and Pattern Recognition. 2024.

**Strengths:**

- The paper proposes a benchmark that uses a diffusion model to evaluate models by reproducing continuous distribution shifts for the first time.

**Additional Feedback:**

There is nothing to mention.

**Clarity:**

Writing is not bad. A structural weakness is that the relationship between motivation and claims has not been explicitly examined.

**Correctness:**

Many of the protocols used to construct the proposed benchmark are heuristics, and correctness cannot be determined because the code and algorithms are not publicly available.

**Documentation:**

There is nothing to mention.

**Ethics:**

There is nothing to mention.

**Limitations:**

Limitation is discussed in the checklist. The paper should also discuss the types of distribution shifts that the proposed benchmark can accommodate.

**Opportunities For Improvement:**

- The papers should publish code to show reproducibility and implementation details.
- The paper should show how well the synthetic dataset reproduces real-world distribution shifts.
- The papers should identify the types of distribution shifts and use cases for which the proposed benchmark works well.

**Relation To Prior Work:**

The existing studies presented in Section 2 are comprehensive.

**Summary And Contributions:**

This paper proposes a new benchmark for evaluating the robustness of deep neural network models.
The motivation of the paper is that some of the existing benchmarks can vary the intensity of distribution shifts, but these variations do not cover the real-world distribution shifts and have little diversity.
To address this issue, the paper introduces a synthetic dataset that continuously controls the intensity of distribution shifts using a text-to-image diffusion model.
Specifically, the paper leverages an existing method that controls the output with a LoRA adapter to generate the distribution shift.
To ensure the realism of the synthetic dataset, the paper generates a new image using text inversion based on an image from the ImageNet dataset.
The paper also constructs a filtering dataset consisting of synthetic data and proposes a heuristic to remove out-of-class (OOC) samples using a classifier trained on this dataset.
The paper evaluates several trained models with the proposed benchmarks, but no remarkable findings seem to be obtained.

---

> ### Author Rebuttal · Authors · 2024-08-17
>
> We thank the reviewer for providing such a detailed review. In addition to the aspects discussed in the global response, we address the remaining points here.
>
> > the paper does not describe concrete formal algorithms or protocols, nor does it provide code, making it difficult to verify reproducibility and factuality.
>
> The supplementary material (Sec. A) contains a [link](https://drive.google.com/drive/folders/1ZTbCwrpedcJ3tGS6U5C4NgnGI4PD1qBH?usp=sharing) to both, our code and data, which we will publish together with the paper (as also stated in the Datasheet in Sec. B).
>
> Generation and filtering are explained in Sec. 3.1 and 4, implementation details in Sec. 5.2 and A.4, metrics in Sec. A.1.2, and labeling in Sec. A.5.1. We will add more formal algorithm descriptions for training, generation, and filtering in the final version.
>
> > it does not provide evidence of the extent to which the evaluation by the synthetic data set with the proposed benchmark reproduces real-world distribution shifts.
>
> We refer to the global response (ii).
>
> > there is little evidence to support why continuous change is important.
>
> We refer to the global response (i).
>
> > it is unclear in what cases the proposed benchmark is effective, as distribution shifts may be caused by non-continuous factors (e.g., style)
>
> We agree that, unlike weather conditions, style is typically seen as a non-continuous shift. However, we observed that style shifts also show a change in appearance at different scales.
> While the exact points of style change in Fig. 8d-f) are hard to pinpoint, the cartoon style shift in Fig. 8b) seems less continuous. Still, some visual properties continue to shift towards a more simplistic cartoon even at later steps.
> Our discussions in l.293-298 also highlight the differences between shifts.
>
> > it is not shown whether the continuity of the intensity of the distribution shift by the generative model is related to the continuity of reality.
>
> Our work is based on the observation that increasing the strength of LoRA sliders enhances the learned concept.
> To address the comment, we calculated the difference in CLIP similarities between two consecutively shifted images $I$ at scale levels $k$ and $k-1$: $D=CLIP_{\Delta,shift}(I_k,I_{k-1})=CLIP(I_k,\text{"in <shift>"})-CLIP(I_{k-1},\text{"in <shift>"})$.
> The CLIP alignment to the shift ($D>0$) increases in 73% of the steps, varying by shift (e.g., 85% for snow, 70% for cartoon).
>
> The averaged difference CLIP shift alignments increase for all shifts. We provide the scores averaged over all shifts of the ImageNet-scale benchmark in the following table:
>
> |Scale|$CLIP_{\Delta,shift}(I_\text{scale},I_0)$|
> |--|--|
> |0.5|+0.13|
> |1.0|+0.65|
> |1.5|+1.86|
> |2.0|+2.93|
> |2.5|+3.78|
>
> In addition to Fig. 3 (right) in the main paper, we also refer to Fig. 3 in the rebuttal PDF for the distribution of the OOD-CV dataset.
>
> > most of the findings in the paper are common to and known from existing studies [a,b,c]
>
> We would like to highlight the findings from our benchmark that have not been sufficiently discussed in the literature:
> - The OOD robustness of a classifier varies across different shifts and severity levels (Fig. 1 and 6 in the rebuttal), supporting the need for benchmarks with multiple real-world shifts and scales.
> - We examined the robustness of a diffusion classifier (see Fig. 12 in the supplementary and Fig. 3 in the rebuttal), demonstrating its inferior robustness to the considered shifts.
>
> We agree that some other findings have been shown in other studies. However, the consistent results with previous benchmarks emphasize that synthetic benchmarks can serve as a scalable source for more systematic evaluations of vision models.
>
> We address the three referenced works as follows:
>
> [a] required hundreds of hours of manual effort for only 12 classes, whereas our benchmark includes 100 classes and can easily be extended using our framework.
>
> [b] considered only simple synthetic corruptions, but we support real-world shifts.
>
> [c] While offering a great strategy to find identify failure cases, the benchmark only considers a discrete set of backgrounds, textures, and materials.
>
> > The paper should show how well the synthetic dataset reproduces real-world distribution shifts.
>
> We refer to the global response (ii).
>
> > The papers should identify the types of distribution shifts and use cases for which the proposed benchmark works well.
>
> "benchmark works well" can be interpreted in different ways. We think the reviewer would like to understand how we ensure that the benchmark corresponds to real-world shifts: We ensured the quality of the shifted images through our OOC filtering strategy.
>
> We provide accuracy drops for all considered shifts in Fig. 6 in the rebuttal PDF, showing that the accuracy drops deviate for different shifts.
>
> We are happy to clarify further if we misunderstood the question.
>
> > The paper should also discuss the types of distribution shifts that the proposed benchmark can accommodate.
>
> See Sec. A.1.3 including Fig. 8 and 9 for a list. In general, our framework can easily be extended to any shift that can be learned through a LoRA adapter.
>
> > A structural weakness is that the relationship between motivation and claims has not been explicitly examined.
>
> We motivated applying multiple scales of a nuisance shift and the relevance of considering failure points. Sec. 5.3 presents results and discussions that support this claim, e.g. in l.260, l.277, and l.293. We will add the valuable comments from the reviews to improve on the clarity.
>
> > The paper also constructs a filtering dataset consisting of synthetic data and proposes a heuristic to remove out-of-class (OOC) samples using a classifier trained on this dataset.
>
> No. The final applied OOC filtering purposefully does not involve classifiers trained with a supervised classification loss. All models involved in the final applied filtering were neither trained on ImageNet nor on or our data.

---

> ### Author Response · Authors · 2024-08-29
> **Happy to discuss more**
>
> Dear Reviewer Hf4n,
>
> Thank you again for your review. We hope our rebuttal has addressed your initial comments. If you still have concerns or further questions, we are more than happy to continue discussions and will do our best to provide thorough responses.
> Once again, we greatly appreciate your time and efforts, and thank you for helping us improve the paper.
>
> Sincerely,
>
> The Authors

---

### Official Review · Reviewer_Y3Jb · 2024-07-27
**Great paper**

**Rating:** 9
**Confidence:** 3
**Correctness:** Yes, to the best of my knowledge.
**Clarity:** Yes

**Review:**

I think this is a great paper with a number of positives. The authors propose a very useful dataset using an elegant new method (LoRA-adapters on SD). Data collection and filtering is done very meticulously. And the model benchmarking is also very comprehensive. Additionally, the paper is very well laid out and presented. Due to these reasons, I highly recommend it for publication.

I have no major complaints about the paper. The only thing I would suggest is including more description about some of the relevant topics, which would really help the uninitiated reader:
- Describing in more detail how LoRA adapters are used to generated continuous nuisance-shifted images.
- More detailed description of how you account for the ImageNet distribution.
I think you have briefly talked about both these things in the paper. But they both rely on existing previous work. I would appreciate if you could include more context from those papers in your text.

**Strengths:**

See "Review"

**Additional Feedback:**

None

**Documentation:**

Yes

**Opportunities For Improvement:**

I don't have many complaints. I have a few suggestions that I have listed in the Review section.

**Relation To Prior Work:**

Yes

**Summary And Contributions:**

Most existing benchmarks measure robustness by evaluating models on a clean image and a corrupted version. Additionally, although there are a few exceptions, most benchmarks evaluate models only on nuisance shifts either already prevalent in natural images or those that are possible to engineer by hand. This paper departs from both traditions in the literature by introducing a dataset of synthetic images (generated using a text-to-image diffusion model) corrupted at continuous set of levels using LoRA adapters. They present a method for ensuring that the generated images are not out-of-class, and then benchmark the robustness of several SoTA models on these images.

---

> ### Author Rebuttal · Authors · 2024-08-17
>
> Thank you for the very positive review and for appreciating the relevance of our work.
>
> > Describing in more detail how LoRA adapters are used to generated continuous nuisance-shifted images.
> More detailed description of how you account for the ImageNet distribution. I think you have briefly talked about both these things in the paper. But they both rely on existing previous work. I would appreciate if you could include more context from those papers in your text.
>
> We will add further explanations on our proposed LoRA-based approach for applying class-specific nuisance shifts in the updated manuscript. Below is a brief overview:
>
> To account for the ImageNet distribution, we use text embeddings [46] that were trained using textual inversion to best capture the image distribution of the considered ImageNet class $c_i$. Generating images using the new additional token `<ImageNet-class-i>` results in images that contain the class-specific characteristics. This method allows us t nearly match the original ImageNet accuracy.
>
> Once these class-specific concepts are learned, we train the LoRA adapters, which are low rank matrices that modify the U-Net parameters $W'=W+s\cdot AB$, where $s$ is the scale and $AB$ is the low rank matrix. The LoRA adapters are trained as follows [14]: Given a class concept $c_t$ and a nuisance concept $c_+$, represented by two text-embeddings `<class>` and `<class> in <shift>`, we optimize the LoRA parameters $\theta_c^*$ using the noise prediction objective in diffusion models: $\text{MSE}(\epsilon_{\theta^*}(X,c_t,t); \epsilon_{\theta}(X,c_t,t)+\epsilon_{\theta}(X,c_+,t))$, where the time steps $t$ are randomly sampled during optimization.
>
> At the end, the trained LoRA adapters capture the direction between the two language concepts via a low-rank approximation, which generalizes to other concepts and images.  We found that learning class-specific LoRA sliders produces higher-quality shifts. Hence, we train separate LoRA adapters for each ImageNet class and shift.
>
> These learned directions can then be applied at various scales $s$ during inference. During sampling, we only apply the LoRA adapters at later steps to avoid significantly altering the image's semantic structure.
> While our benchmark uses 5 scales, the framework and learned adapters can be applied at any scale. However, slider extrapolation ($s>1$) fails at certain scales, which may vary by class and seed. For this reason, we apply an out-of-class filtering strategy.
>
> ---
>
> **References**
>
> [14] R. Gandikota et al. Concept sliders: LoRA adaptors for precise control in diffusion models, 2023.
>
> [46] J. Vendrow et al. Dataset interfaces: Diagnosing model failures using controllable counterfactual generation, 2023.

---

> ### Author Response · Authors · 2024-08-29
> **Happy to discuss more**
>
> Dear Reviewer Y3Jb,
>
> Thank you again for your review. We hope our rebuttal has addressed your initial comments. If you have other concerns or further questions, we are more than happy to continue discussions and will do our best to provide thorough responses.
> Once again, we greatly appreciate your time and efforts, and thank you for helping us improve the paper.
>
> Sincerely,
>
> The Authors

---

### Official Review · Reviewer_nzJ3 · 2024-07-31

**Rating:** 6
**Confidence:** 3

**Review:**

**Strengths.**

Quality: The benchmark construction is mostly sound, and the authors put considerable work into analyzing the effectiveness of their synthetic generation and filtering method to produce the evaluation data. The experiments are also performed at a large enough scale to mitigate concerns about error bars, and experiments are run across a variety of different models and settings.

Clarity: The filtering procedures and all findings are enumerated and explained in detail.

Originality: While the use of text-to-image models to produce a continuous scale of variations to an image is not novel, the paper suggests a strong filtering method and provides extensive experiments analyzing the robustness of vision models to such variations.


**Questions/Opportunities for Improvement.**

Questions:
1. Does the generation and/or filtering method bias towards or against certain models? For example, since SDv2.0 uses OpenCLIP trained on LAION, does that make it more likely that CLIP-LAION would naturally have a smaller accuracy drop? This wouldn't necessarily invalidate results, but would probably preclude fair comparisons against certain models.
2. What exactly is signified by the nuisance scale (on the X axes)? If it refers to the strength of the LoRA adapter, then wouldn't we expect a scale of 1.0 to roughly correspond to the same distribution as the 2P method (Figure 3)?
3. It would be nice to see a plot of original accuracy vs accuracy drop for each nuisance scale, as a potential common confounder to the factors that were ablated like model size and architecture.
4. How much of the cumulative failure point rates are accounted for by noise or randomness? For example, in a general classification case, if you add a fixed amount of Gaussian noise to the images in the test set, then we should expect the accuracy drop to be roughly the same across different seeds, but the cumulative number of "failure points" would be higher because different instances would be correct and incorrect across the different seeds. Since the nuisance generation method results in a small but still significant change to the input image, one might expect a similar baseline amount of failure points.

Significance: It remains unclear how significant the contributions of the paper are. While there are several interesting results from the experiments, it would help to explore them in more detail and perhaps even compare to findings in other papers. Do "modern architectures improve the robustness" and "larger models are more robust" both actually correlate with model base accuracy? Or do the architectures or sizes actually specifically provide advantages to this form of robustness? What does it imply when "some models have a larger accuracy drop but fail later"? Overall, it is also not very clear what further advantages and insights, the new benchmark provides, i.e., what would researchers learn about new vision models that could not be gained from existing benchmarks?

**Strengths:**

See "Review" section.

**Additional Feedback:**

None.

**Clarity:**

The paper is mostly well-written, though section 5 could be better organized to separate out experimental details from results.

**Correctness:**

The dataset and benchmark are constructed and executed correctly, with human verification of data filtering methods and extensive experiments on the benchmark side.

**Documentation:**

All required materials and details are provided in the paper or supplementary.

**Ethics:**

I do not suspect any ethical concerns with the submission.

**Limitations:**

The authors mostly address potential limitations of their dataset creation. One loose end is addressing how applicable this evaluation of robustness is: will accuracy drops on synthetically heavily altered images typically correlate or be relevant to performance on real-world images and applications?

**Opportunities For Improvement:**

See "Review" section.

**Relation To Prior Work:**

The Related Works section is sufficient.

**Summary And Contributions:**

The paper evaluates the robustness of vision models in classification under continuous nuisance shifts. To obtain images displaying a continuous range of nuisance factors, the authors apply LoRA adapters at varying strengths to a base Stable Diffusion text-to-image model to produce images of the target class with the desired alteration. The benchmark is constructed around a random subset of 100 ImageNet classes, from which a total of 24k images are generated with varying nuisances and nuisance scales. To ensure that the synthetic images still correspond to the gold label class, images are filtered with a combination of CLIP similarity between generated image and class text, and CLIP similarity between generated image and original image.

The paper evaluates models across model architecture, model size, and training paradigm, demonstrates that larger models and broader pretraining data improve continuous nuisance shift robustness, and shows that accuracy drop and failure points (nuisance scale at which the model starts to make errors) are not exactly correlated. It suggests that failure point analysis should be considered as a separate axis when measuring model robustness.

---

> ### Author Rebuttal · Authors · 2024-08-17
>
> > Does the generation and/or filtering method bias towards or against certain models?
>
> Yes, we agree that the generation method and filtering models are likely biased by CLIP and DINOv2 training data. Therefore, we avoid direct comparisons between models trained on CLIP or DINOv2 data and those that are not.
>
> We reduce some of the biases or correlations from CLIP in the generation process by
>
> - training LoRA adapters that learn concepts from CLIP but express them differently than in text-embedding space and
> - using the SDEdit formulation [33], which adds flexibility in applying the nuisance shift (see Fig. 2 in the main paper for an example).
>
> To reduce bias in the accuracy estimate,
>
> - our filtering mechanism combines multiple models trained on different datasets that do not include ImageNet.
> - Additionally, we calibrate the filtering mechanism using a manually labeled dataset to reduce biases.
>
>
> > What exactly is signified by the nuisance scale (on the X axes)? If it refers to the strength of the LoRA adapter, then wouldn't we expect a scale of 1.0 to roughly correspond to the same distribution as the 2P method (Figure 3)?
>
> Yes, the scale refers to the strength of the LoRA adapters. We also considered other x-axes, such as CLIP alignment to the shift. However, we chose the LoRA scale because it directly relates to the LoRA slider, where a larger scale corresponds to a stronger application of the learned nuisance shift. We will clarify this choice in the updated manuscript.
>
>
> In general, a scale of 1 would be expected to result in a similar CLIP alignment to the 2P scenario on average if all noise steps were active. Following [33], we selected a lower number of active noise steps to reduce the effect of other variations, such as change in semantic structure. This approach also led to a more stable image editing.
>
>
> > It would be nice to see a plot of original accuracy vs accuracy drop for each nuisance scale, as a potential common confounder to the factors that were ablated like model size and architecture.
> > Do "modern architectures improve the robustness" and "larger models are more robust" both actually correlate with model base accuracy? Or do the architectures or sizes actually specifically provide advantages to this form of robustness?
>
> Thanks a lot for pointing out this consideration. While [55] claims that the base accuracy correlates with OOD performance, [21] contradicts this. Therefore, we focused on the accuracy drop as a measure of OOD robustness.
>
> Nonetheless, we explored this relationship in our large-scale study of 40 models. Fig. 4 in the rebuttal PDF shows that while base accuracy is not correlated with some shifts, it is correlated with others. This provides a more nuanced view compared to previous studies [55,21]: The correlation between base accuracy and OOD robustness depends on the _type_ of distribution shift. For example, shifts caused by corruptions (e.g., weather changes) are correlated with base accuracy, but style distribution shifts do not appear to have a statistically significant correlation.
>
> > How much of the cumulative failure point rates are accounted for by noise or randomness?
>
> Thank you for raising this question. Different seeds lead to different generated images whereas for a given seed and scale the resulting image generation is deterministic.
> However, for a fixed seed, we find that the LoRA adapter consistently increases the strength of the learned concept (as measured by CLIP score).
> The failure point metric is based on the first failing scale, comparing models on individual starting images to determine when the model begins to fail. If a model correctly classified a larger scale after having failed for a smaller scale, the cumulative number of failure points would be higher than the accuracy drop.
>
> In response to the reviewer's question, we examined whether our statements related to the failure point results in Sec. 5.3 were statistically significant. We applied Mood's median test with a p-level of 0.05 and found that all related statements (l.260-265, l.268, l.277) are statistically significant.
>
> We are not entirely sure if we understood this question correctly and are happy to clarify further if we did not address this point accurately.
>
> > What does it imply when "some models have a larger accuracy drop but fail later"?
>
> We thank the reviewer for highlighting that unclear statement. We will revise it for clarity.
> If model A has a larger average accuracy drop than model B, it could indicate that model A generally performs better but suddenly fails completely at the largest scale, which might be too extreme for its intended application.
> This scenario can only be captured by considering various scales during benchmarking.
>
> > Overall, it is also not very clear what further advantages and insights, the new benchmark provides, i.e., what would researchers learn about new vision models that could not be gained from existing benchmarks?
>
> We addressed this question in the global response (i).
>
> > Will accuracy drops on synthetically heavily altered images typically correlate or be relevant to performance on real-world images and applications?
>
> We addressed this question in the global response (ii).
>
> > The paper is mostly well-written, though section 5 could be better organized to separate out experimental details from results.
>
> We will move the mentioned results in the experimental details to the main results section.
>
> ---
>
> **References**
>
> [21] D. Hendrycks et al. The many faces of robustness: A critical analysis of out-of-distribution generalization. In ICCV, 2021.
>
> [33] C. Meng et al. Sdedit: Guided image synthesis and editing with stochastic differential equations. In ICLR, 2021.
>
> [55] R. Taori et al. Measuring Robustness to Natural Distribution Shifts in Image Classification. In NeurIPS, 2020.

---

> > ### Comment · Reviewer_nzJ3 · 2024-08-30
> > **Response to Rebuttal**
> >
> > Thank you for thoroughly answering my questions, as well as for backing up your claims to the significance of your experiments and results. I will increase my score.
> >
> > Regarding:
> > > We are not entirely sure if we understood this question correctly and are happy to clarify further if we did not address this point accurately.
> >
> > To clarify my original question:
> >
> > > How much of the cumulative failure point rates are accounted for by noise or randomness?
> >
> > As you mentioned, fixing both parameters (seed and LoRA scale) results in a consistent image. However, if the LoRA scale is changed, even with seed held constant, the generated image will change. In this setting, I would expect the performance of the classifier to be somewhat random (since at each scale there is, obviously, a different image). Even if increasing the LoRA scale didn't make the images harder, we would still see some baseline number of failure points because of randomness. So my question was whether that baseline expected number of failure points was taken into account in the analysis.
> >
> > That being said, I believe the statistical significance test you performed clears doubts about that question, so thank you for running that analysis.

---

> ### Author Response · Authors · 2024-08-29
> **Happy to discuss more**
>
> Dear Reviewer nzJ3,
>
> Thank you again for your review. We hope our rebuttal has addressed your initial comments. If you still have any concerns or further questions, we are more than happy to continue discussions and will do our best to provide thorough responses.
> Once again, we greatly appreciate your time and efforts, and thank you for helping us improve the paper.
>
> Sincerely,
>
> The Authors

---

### Author Rebuttal · Authors · 2024-08-17

We thank all reviewers for their raised questions and constructive feedback. We are pleased that the reviewers recognize the usefulness (RY3Jb) and originality (RHf4n) of our approach, the meticulous data collection (RY3Jb) and strong out-of-class filtering method (RnzJ3), and the sound construction (RnzJ3) and comprehensiveness of our benchmark (RY3Jb).

In the following, we discuss the common questions among the reviewers.

> (i) Limited **significance** of benchmarking various nuisance shift levels. (RnzJ3,RHf4n)

Evaluating shifts at various scales is relevant for benchmarking for the three following reasons:

**1) Robustness as a spectrum.**

Various works point out that robustness should not be reduced to a single metric [9,21]. In contrast, it should be considered as a spectrum of various metrics, each specifically addressing robustness in relation to a considered distribution shift.

**2) Classifier performance varies for different scales.**

As noted by [20] in the construction of ImageNet-C, nuisance shifts affect classifier performance differently depending on their severity. Evaluating performance across multiple shift levels gives a more complete picture of robustness.
Additionally, [9] highlights that averaging performance over severity levels, as done in [20], can remove key insights since it does not reflect the sensitivity of a model to specific corruptions or shifts. Our benchmark aligns with this, enabling evaluation across various real-world shifts beyond simple corruptions, while remaining scalable.

**In our benchmark, we observe that model performance rankings change across different nuisance shift levels.**
We demonstrate this using the model axis and painting style as an example below:

|scale|RN152|ViT/B|DeiT/B|DeiT3/B|CN/B|
|--|--|--|--|--|--|
|0|1|1|1|1|1|
|0.5|4|3|5|1|2|
|1|2|4|5|3|1|
|1.5|5|1|2|4|3|
|2|4|5|1|2|3|
|2.5|3|5|4|1|2|

We refer to the accuracy drops in Fig. 1b) in the rebuttal PDF.

Similarly, to further support this argument, we use 7 levels of contrast as a deterministic example corruption from ImageNet-C, based on the implementation of [20]. As shown in Tab. 1 of the rebuttal, model rankings shift across different levels. A global averaged metric fails to capture these variations effectively.


**3) Relevance of the failure point of a vision model in downstream applications.**

In certain applications, it is important to know the expected performance at specific nuisance shift levels, rather than just a global accuracy drop. For instance, an autonomous driving company may need to determine the fog density at which system performance falls below a critical threshold.

We will include this motivation in the updated manuscript.

> (ii) Are synthetic distribution shifts realistic? (RnzJ3,RHf4n)

The **OOD-CV dataset** [52] includes various annotated real-world nuisance shifts. We specifically use this dataset to compare our synthetic nuisance shifts with real-world weather shifts before scaling the analysis to ImageNet.

Our results show that synthetic data leads to a similar or lower accuracy drop compared to real-world images with the same nuisance shift (see Fig. 5 (left) in the main paper). These results were manually checked to ensure the presence of the desired shift and object, confirming the realism of the implemented shift. The accuracy drop gap can be explained by the fact that the real-world images in OOD-CV are entangled with other shifts or corruptions, as discussed in Sec. 5.1 (l.213-221) and Fig. 17.

We visually demonstrate that our applied shifts capture real-world nuisance shifts from the OOD-CV dataset in Fig. 2 of the rebuttal PDF. To measure the correlation of our implemented shifts, we show the CLIP alignment to various shift levels in Fig. 3 of the rebuttal PDF, similar to Fig. 3 (right) in the main paper.

To ensure the images align with the desired class, we applied the **human labeling strategy and out-of-class filtering pipeline**.

Additionally, we **fine-tuned a ResNet-50 classifier using our synthetic data**. The results indicate improved performance on the shifted real-world dataset, without a significant decline on the original ImageNet dataset.
We present the results comparing a non-fine-tuned model with a model fine-tuned using 50% synthetic data and 50% ImageNet training data:

| Evaluation Dataset|wo/ fine-tuning|w/ fine-tuning|
|:-|-:|-:|
| IN/val|80.15|78.11|
| IN/R|27.34|37.57|

---

**References**

[9] N. Drenkow et al. Robustness in deep learning for computer vision: Mind the gap? CoRR, abs/2112.00639, 2021.

[20] D. Hendrycks et al. Benchmarking neural network robustness to common corruptions and perturbations. In ICLR, 2019.

[21] D. Hendrycks et al. The many faces of robustness: A critical analysis of out-of-distribution generalization. In ICCV, 2021.

[52] B. Zhao et al. Ood-cv: A benchmark for robustness to out-of-distribution shifts of individual nuisances in natural images. In ECCV, 2022.

---

### Author Response · Authors · 2024-08-27
**Looking forward to further discussions**

Dear Reviewers,

Thank you for your valuable time and constructive feedback.

In response to your concerns, we have provided additional discussions, technical details, or experimental results. We hope these clarifications address the raised points and provide greater clarity.
We will incorporate several suggested changes in our revision, with the goal of improving the quality and impact of our work.

There are, however, a few open questions that remain. We would greatly appreciate your opinions to help us ensure that our responses addressed your concerns.

Thank you again for your time. We are more than willing to provide additional clarifications if there are any other questions about our work.

---

### Comment · Area_Chair_tSNJ · 2024-08-29
**Discussion period ends soon**

Dear reviewers,

Thank you for taking the time and effort in providing reviews. The authors have provided detailed responses to many of your comments. The author-reviewer discussion period will end soon (August 31, 2024), so please take some time to go through their responses and discuss with the authors, as this is an important part of the peer review process. Additionally, please update your scores, along with an explanation for any adjustments.

---

### Decision · Program_Chairs · 2024-09-26

**Decision:**

Reject

**Comment:**

This paper introduces a method to evaluate vision model robustness in a more controlled fashion, using generative models to build a continuous robustness benchmark for varying nuisance shifts. Most reviewers agree the paper is well written, and presents a useful dataset for evaluating robustness. One reviewer raised concerns about reproducibility, but the authors have provided a link to code and data and promised to add more formal descriptions of their data generation in the final version. Given this, we believe this paper clears the bar for acceptance at NeurIPS Datasets and Benchmarks.

We encourage the authors to address the remaining review feedback in the final version. In particular, an underlying concern is the importance of this benchmark compared to other robustness benchmarks. The author response in the general rebuttal [part (i)] is a good start, and we encourage authors to incorporate it into the final version of the paper.

Note from PC: This year, the track has been incredibly competitive, which meant that many good papers had to be rejected. After careful discussion with the SACs we have concluded that this paper unfortunately cannot be accepted this time. This is the final decision, which cannot be appealed. We encourage the authors to incorporate feedback from reviewers and additional results / discussion provided during the author response period in their next submission.